



# Investigating the relationship between volume transport and sea surface height in a numerical ocean model

Estee Vermeulen [1,2*], Björn Backeberg [2,3,4], Juliet Hermes [1,5], Shane Elipot [6]

[1] Department of Oceanography, University of Cape Town, Rondebosch, South Africa
[2] Nansen-Tutu Centre for Marine Environmental Research, University of Cape Town, South Africa
[3] CSIR, Coastal Systems Research Group, Stellenbosch, South Africa
[4] Nansen Environmental and Remote Sensing Centre, Bergen, Norway
[5] South African Environmental Observation Network, Egagasini Node, Cape Town, South Africa
[6] Rosenstiel School of Marine and Atmospheric Science, University of Miami, 4600 Rickenbacker Causeway, Miami, FL 33149

[*]*Corresponding author address:* Estee Vermeulen, Department of Oceanography, University of Cape Town, Rondebosch, South Africa
Email: esteever01@gmail.com





## 17 Abstract

The Agulhas Current Time-series mooring array (ACT) measured transport of the Agulhas
Current at 34°S for a period of 3 years. Using along-track satellite altimetry data directly
above the array, a proxy of Agulhas Current transport was developed based on the relationship
between cross-current sea surface height (SSH) gradients and the measured transports. In this
study, the robustness of the proxy is tested within a numerical modelling framework, using a
34-year long regional-hindcast simulation from the Hybrid Coordinate Ocean Model (HYCOM).
Two reference proxies were created using HYCOM data from 2010-2013, extracting model data
at the mooring positions and along the satellite altimeter track for; (1) the box transport
($T_{\mathrm{box}}$) and (2) the jet (southwestward) transport ($T_{jet}$). Next, sensitivity tests were performed
where the proxy was recalculated from HYCOM for (1) a period where the modelled vertical
stratification was different compared to the reference proxy, and (2) different lengths of periods:
1, 3, 6, 12, 18 and 34 years. Compared to the simulated (native) transports, it was found that
the HYCOM proxy was more capable of estimating the box transport of the Agulhas Current
compared to the jet transport. The HYCOM configuration in this study contained exaggerated
levels of offshore variability in the form of frequently-impinging baroclinic anticyclonic eddies.
These eddies consequently broke down the linear relationship between SSH slope and vertically-
integrated transport, resulting in stronger correlations for the inshore linear regression models
compared to the ones offshore. Vertically-integrated transport estimates were therefore more
accurate inshore than those offshore or when the current was in a meandering state. Results
showed that calculating the proxy over shorter or longer time periods in the model did not
significantly impact the skill of the Agulhas transport proxy, suggesting that 3-years was a
sufficiently long time-period for the observation based transport proxy.



## 1 Introduction

The Agulhas Current System is the strongest western boundary current in the Southern Hemisphere and transports warm tropical water southward along the east coast of South Africa [Lutjeharms, 2006]. The Agulhas Current, in the northern region, is known for its narrow, fast, flow conditions following the steep continental slope [de Ruijter et al., 1999]. As the current continues southwestward the current becomes increasingly unstable over the widening continental shelf until it eventually retroflects, forming an anticyclonic loop south of Africa and returning to the Indian Ocean as the eastward Agulhas Return Current [Beal et al., 2011; Biastoch and Krauss, 1999; Dijkstra and de Ruijter, 2001; Hermes et al., 2007; Lutjeharms, 2006; Loveday et al., 2014]. The anticyclonic loop, known as the Agulhas retroflection, contains some of the highest levels of mesoscale variability in the global ocean [Gordon, 2003] through the formation of Agulhas rings, eddies and filaments. This, in turn, contributes significantly to the Benguela upwelling system, the Atlantic Ocean and the global overturning circulation system [Gordon et al., 1987; Beal et al., 2011; Durgadoo et al., 2013], thereby impacting the Atlantic Meridional Overturning Circulation (AMOC) by providing a salt-advective feedback through a process known as the Agulhas leakage [Biastoch and Krauss, 1999; Beal et al., 2011; Durgadoo et al., 2013; Loveday et al., 2014]. In the regional context, the Agulhas Current has a major influence on the local weather systems, due to large latent and sensible heat fluxes, which contributes to rainfall and storm events over the adjacent land [Reason, 2001; Rouault et al., 2002; Rouault and Lutjeharms, 2003]. The unique circulation of the Agulhas Current System, in the context of regional and global climates, makes it an important field of research.

To understand the complicated dynamics of the Agulhas Current requires an integrated approach using numerical ocean models, satellite remote sensing measurements and *in situ* observations. Previous studies have suggested that measuring the dynamics of the Agulhas Current in the northern region is easier due to its stable trajectory and its confinement to the continental slope [van Sebille et al., 2010]. However, the close proximity of the current to the coast makes it difficult to monitor using satellite altimetry [Rouault et al., 2010]. In addition, the frequent disturbances of the current in the form of solitary meanders, also



known as Natal Pulses, and its interactions with mesoscale features originating upstream
and from the east [Elipot and Beal, 2015], remain poorly resolved in many numerical
ocean models [Tsugawa and Hasumi, 2010; Braby et al., 2016], highlighting the challenges
involved in monitoring and modelling the dynamics in this region.
There is a trade-off between spatial and temporal sampling. *In situ* observations may
accurately measure the dynamics of the Agulhas Current throughout the water column
but are expensive and spatially coarse. In contrast, satellite observations can provide
high spatial resolution data of the surface ocean but lacks detailed information below
the surface. Hence, numerical models are needed to provide a temporally coherent, high
resolution representation of the ocean throughout the water column. Numerous studies
aiming to monitor long-term changes in global current systems have adopted methods
to combine various sampling tools [eg. Maul et al. 1990; Imawaki et al. 2001; Andres
et al. 2008; Zhu et al. 2004; Yan and Sun 2015], including the recent development of the
Agulhas transport proxy established to monitor the interannual variability and long-term
trends in Agulhas Current transport [Beal and Elipot, 2016].
The Agulhas transport proxy of Beal and Elipot [2016] was derived from the physical
principle of geostrophy, where along-track sea surface height slope measured by satellite
altimeters can ultimately be related to a measure of volume transport across a portion
of the current, provided that the surface current represents the flow at depth [Beal and
Elipot, 2016]. Beal and Elipot [2016] have shown that a strong relationship exists between
surface geostrophic velocity and full-depth transport such that sea level anomalies can
be used to study the variability and dynamics of the Agulhas Current System as has
been demonstrated before [Fu et al., 2010; Rouault et al., 2010; Rouault and Penven,
2011; etc.]. The 22-year transport proxy created by Beal and Elipot [2016] assumed
a fixed linear relationship between *in situ* transport and sea surface slope based on *in*
*situ* measurements over the 3-year sampling period of the Agulhas Current Time-series
experiment (ACT) [Beal et al., 2015]. Analyses of the Agulhas Current transport proxy
time-series concluded that the Agulhas Current has not intensified over the last two
decades in response to intensified global winds under anthropogenic climate change [Cai,
2006; Yang et al., 2016], but instead has broadened as a result of increased eddy activity
[Beal and Elipot, 2016] in agreement with Backeberg et al. [2012].





This modelling study aimed to recreate the Agulhas transport proxy developed by Beal
and Elipot [2016], within a regional HYCOM simulation of the greater Agulhas Current
System in order to test the validity of the underlying assumptions on which the satellite-
altimeter derived proxy was based. Firstly, the Agulhas Current transport proxy was
recreated using modelled data from HYCOM following the methodology of Beal et al.
[2015] and [Beal and Elipot, 2016] for the data period 2010-2013. This reference proxy
allowed for the relationship between Agulhas Current transports and sea surface slope
across the Agulhas Current Time-series experiment (ACT) array to be investigated in
HYCOM. Following this, the impact of the vertical variability of the current on the accur-
acy of the transport proxy was assessed. Finally, the optimal length scale of observations
needed to build a strong linear relationship between transport and SSH slope was tested
by recalculating the proxy using 1, 3, 6, 12, 18 and 34 years of HYCOM data.
Assuming a constant vertical stratification over the 3-year sampling period, and hence
ignoring baroclinic changes that could potentially impact the linear relationship between
sea surface slope and full-depth transport could become problematic when generating a
22-year proxy of Agulhas Current transports. Therefore, key questions for this paper in-
clude: (1) How is the linear relationship between transport and sea surface slope affected
when recalculating the proxy over longer time-periods in HYCOM? (2) How will changes
in the vertical structure of the Agulhas Current impact the transport proxy? Theoret-
ically the vertical velocity structure changes during mesoscale meander events Zhu et al.
[2004] and thermohaline processes [Beal and Elipot, 2016] since horizontal changes in
stratification result in changes in the velocity structure with depth. Perhaps even changes
in the strength of the Agulhas Undercurrent may impact the transport proxy. Finally,
(3) what would be the ideal sampling period needed to build a strong, linear relationship
between transport and SSH slope? Building the linear relationship over periods longer
than 3 years could perhaps increase the skill of the transport proxy by averaging out
random perturbations, but may be also be affected by the interannual variability of the
current system [Elipot and Beal, 2018]. This study aims to test the robustness of using 3
years of *in situ* mooring data to develop a satellite altimetry derived transport proxy for
the Agulhas Current at 34°S, by testing the underlying assumptions in a numerical mod-
elling framework. This can assist in planning future deployments of moorings ultimately


<sub>132</sub> facilitating the improvement of an integrated ocean observing system for the Agulhas

<sub>133</sub> Current.

<sub>134</sub> This paper is structured as follows; Section 2 describes the data and methods, it should

<sub>135</sub> be noted that this section forms a key part of the paper as the methods of recreating the

<sub>136</sub> proxy are an integral component of the study. Section 3 presents the results from the

<sub>137</sub> HYCOM transport proxy and lastly Section 4 presents the summary and conclusions.

## 2 Data and Methods

### 2.1 The Hybrid Coordinate Ocean Model

<sub>140</sub> The Hybrid Coordinate Ocean Model (HYCOM) is a primitive equation ocean model

<sub>141</sub> that was developed from the Miami Isopycnic Coordinate Ocean Model (MICOM) [Smith

<sub>142</sub> et al., 1990]. HYCOM combines the optimal features of isopycnic-coordinate and fixed-

<sub>143</sub> grid ocean circulation models into one framework [Bleck, 2002] and uses the hybrid layers

<sub>144</sub> to change the vertical coordinates depending on the stratification of the water column.

<sub>145</sub> The model makes a dynamically smooth transition between the vertical coordinate types

<sub>146</sub> via the continuity equation using the hybrid coordinate generator [Chassignet et al., 2007].

<sub>147</sub> Well-mixed surface layers use z-level coordinates, $\rho$-coordinates are utilized between the

<sub>148</sub> surface and bottom layers in a well-stratified ocean, and the bottom layers apply $\sigma$-

<sub>149</sub> coordinates following bottom topography. Adjusting the vertical spacing between the

<sub>150</sub> hybrid coordinate layers in HYCOM simplifies the numerical implementation of several

<sub>151</sub> physical processes without affecting the efficient vertical resolution, and in doing so com-

<sub>152</sub> bines the advantages of the different coordinate types in optimally simulating coastal and

<sub>153</sub> open-ocean circulation features [Chassignet et al., 2007].

<sub>154</sub> The HYCOM output in this study was made available from a nested 1/10° model of

<sub>155</sub> the greater Agulhas Current System (AGULHAS) [Backeberg et al., 2008; 2009; 2014].

<sub>156</sub> The regional nested model, AGULHAS, received boundary conditions from the basin-

<sub>157</sub> scale model of the Indian and Southern Ocean (INDIA) [George et al., 2010] every 6-hrs.

<sub>158</sub> The boundary conditions were relaxed towards the outer model over a 20 grid cell buffer

<sub>159</sub> zone. The horizontal resolution of the parent model ranged from 14 km in the northern

<sub>160</sub> Indian Ocean to 45 km in the Southern Ocean, with a resolution ranging from 30 to 40





km in the region of the Agulhas Current. The nested model covered the region from
the Mozambique Channel to the Agulhas Retroflection region and the Agulhas Return
Current, geographically extending from approximately 0°-60° East and from 10°-50° South,
with a horizontal resolution of ∼10 km that adequately resolved mesoscale dynamics to
the order of the first baroclinic Rossby radius estimated to be about 30 km [Chelton et al.,
1998]. Both models have 30 hybrid layers and targeted densities ranging from 23.6 to 27.6
kg/m$^3$.
The parent model was initialised from Levitus climatology (WOA05) [Antonov and Levi-
tus, 2006] and spun up for 10 years using climatological ERA-interim forcing [Dee et al.,
2011]. AGULHAS was initialised from a balanced field of the parent model interpolated
to the high-resolution grid. Both models were then run from 1980 to 2014 using inter-
annual forcing from ERA40 [Uppala et al., 2005] and ERA-interim [Dee et al., 2011].
Version 2.2 of the HYCOM source code has been used in this model and, together with
the second order advection scheme, provides an adequate representation of the Agulhas
Current [Backeberg et al., 2014]. However, limitations of the free running model include
high levels of SSH variability south of Madagascar and offshore of the Agulhas Current,
suggesting that eddy trajectories may be too regular in the model [Backeberg et al.,
2014]. The data available for this study was a weekly output of the regional HYCOM
model of the Agulhas region from 1980 to 2014. See table 1 for a summary of the model
configuration.

## 2.2 The Agulhas Current Time-series Experiment

The ACT experiment was established to notably obtain a multi-decadal proxy of Agulhas
Current transport using satellite altimeter data. The first phase of the experiment was
the *in situ* phase where the ACT mooring array was deployed in the Agulhas Current,
near 34°S, for a period of three years from 2010-2013 [Beal et al., 2015]. The second
phase was the development of the transport proxy, where sea surface height along the
ACT section, obtained from along-track satellite altimetry, was regressed to the *in situ*
transport measurements [van Sebille et al., 2010; Beal and Elipot, 2016]. To optimally fa-
cilitate the regression between the transport and altimetry, the ACT array was collocated
with the altimeter track number 96 successively occupied by satellites TOPEX/Poseidon





Table 1: HYCOM specifications.

| Model | HYCOM (regional) |
|---|---|
| Configuration | AGULHAS (nested) |
| Nested domain | 0°-60°E; 10°-50°S |
| Time period | 1980-2014 |
| Resolution | 1/10°; Weekly (7/8 days) |
| Grid spacing (km) | ~10 km |
| Vertical discretization | 30 hybrid layers<br>Target densities (+1,000 kg/m3)<br>layer 1 - layer 30:<br>22.30, 22.60, 22.90, 23.20, 23.50,<br>23.80, 24.10, 24.40, 24.70, 25.00,<br>25.30, 25.60, 25.90, 26.20, 26.50,<br>26.80, 26.89, 26.99, 27.08, 27.18,<br>27.27, 27.37, 27.46, 27.56, 27.65,<br>27.75, 27.84, 27.94, 28.00, 28.05 |
| Bathymetry | GEBCO 1' |
| Atmospheric forcing | 6-hourly ERA-interim reanalysis data (1/4°) resolution |
| Boundary forcing | Parent model (INDIA) |
| Advection scheme | $2^{nd}$ order |
| Vertical mixing scheme | KPP |

(1992-2002), Jason-1 (2002-2008) and currently Jason-2 (since 2008) and Jason-3 (since
2016) [Beal and Elipot, 2016] (Figure 1).
During the first phase of the ACT experiment, the mooring array was maintained in the
Agulhas Current for a period of 34 months, perpendicular to the continental slope at
34°S, south of East London, South Africa (Figure 1). The array was made up of 12 sites;
site A through G were full-depth current meter moorings which were, on average, 26 km
apart. Sites P2-P5 were CPIES (Current- and Pressure-recording Inverted Echo Sounders)
placed 50 km apart. The CPIES were used to estimate the geostrophic cross-track velocity
beyond mooring G so that the Agulhas Current variability was fully-captured during





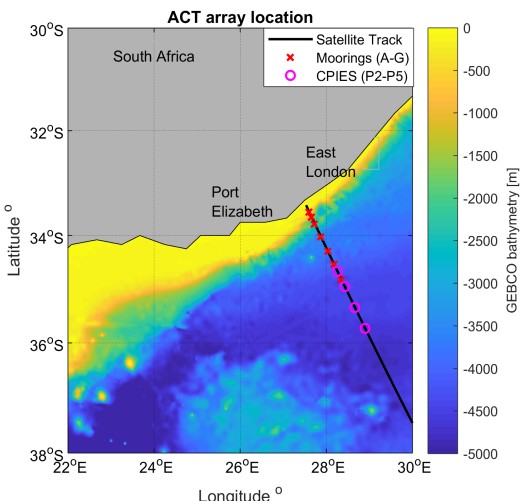

Figure 1: Geographical location of the ACT array with the mooring (red crosses) and CPIES (magenta circles) stations relative to the T/P, Jason-1,2,3 satellite track #96 (black line). Colour shading illustrates the GEBCO bathymetry (m).

meander events [Beal et al., 2015]. From the data collected in Beal et al. [2015], two volume transports were estimated: (1) a box or boundary layer transport ($T_{box}$) and (2) a western boundary jet transport ($T_{jet}$). $T_{box}$ is a net transport within a fixed distance from the coast, while $T_{jet}$ is a stream dependent transport that is calculated by changing the boundaries of integration at each time step depending on the strength and cross-sectional area of the southwestward jet. The western boundary jet transport algorithm was developed to specifically exclude the northeastward transport during meander events, occurring inshore of the meander [Beal et al., 2015].

During the second phase of the ACT experiment, Beal and Elipot [2016] built a 22-year transport proxy by regressing the three years of *in situ* transport measurements against along-track satellite altimeter data spanning the years 1993-2015. Beal and Elipot [2016] noted the importance of the relationship between sea surface height and transport when inferring trends in the current structure based on satellite altimetry and remained cautious regarding the assumptions used to validate the proxy. In order to obtain transport estimates using altimetry, it was also important to define accurate boundaries for the Agulhas Current to distinguish whether the current is stable or meandering and to determine the width of the current to calculate $T_{box}$ and $T_{jet}$.



## 2.3 Development of the Agulhas transport proxy

Based on physical principles sea surface slope is proportional to surface geostrophic velocity. Previous analyses have shown that the vertical structure of the Agulhas Current is barotropic [Elipot and Beal, 2015], such that the direction of current velocity anomalies does not change significantly with depth. This suggests that the relationship between surface geostrophic velocity and full depth transport should be strong, despite the presence of the Agulhas Undercurrent [Beal and Elipot, 2016]. The relationship between sea surface slope and transport was therefore tested using linear regression models, which explicitly described a relationship between the predictor variable, sea surface slope and the response variable, transport per unit distance [van Sebille et al., 2010; Beal and Elipot, 2016].

The transport proxy created by Beal and Elipot [2016] was initially developed by finding a linear relationship between transport and sea surface slope across the entire length of the ACT array, a common method used in previous studies [Imawaki et al., 2001; van Sebille et al., 2010; Sprintall and Revelard, 2014; Yan and Sun, 2015]. However, this method lead to uncertainty in the linear regression due to the strong, co-varying sea surface height across the current. The preferred method was therefore to build nine individual linear regression models, one for each mooring position and CPIES-pairs along the ACT array, which locally related transport to sea surface slope [Beal and Elipot, 2016]. It is important to note that the regression models assumed a constant, linear relationship between sea surface slope and transport over the three-year *in situ* period. The transport variable in the regression models was defined as transport per unit distance ($Tx$ and $Txsw$), i.e. the vertically integrated velocity with units in $m^2s^{-1}$. The total transports, $T_{box}$ and $T_{jet}$ in $m^3s^{-1}$, were calculated by integrating the $Tx$ and $Txsw$ estimates, predicted from the regression models, to the respective current boundaries.

## 2.4 Recreating the Agulhas transport proxy in HYCOM

### 2.4.1 Model Transport

In order to recreate the Agulhas Current proxy in HYCOM, data corresponding to the measurements collected from the ACT mooring array were extracted from the model.




The barotropic velocity -equivalent to an integral of the velocity with depth- from each
mooring location (A-G) and CPIES pairs P3-P4 and P4-P5 was extracted for the 34-year
model period. Extracting the barotropic velocity component from each mooring avoided
interpolation errors that may have occurred if the model velocity was interpolated onto
the locations of each current-meter instrument on each mooring [e.g. van Sebille et al.,
2010]. Transport per unit distance ($Tx$) for each mooring was calculated by multiplying
the cross-track barotropic velocity by the respective depth at each mooring location and
the sea surface slope for each of the locations were obtained from the model (see next
section) (hereafter CPIES pairs P3-P4 and P4-P5 were included as mooring positions 8
and 9). The same method was employed to build regression models between sea surface
slope and the southwestward component of the flow ($Txsw$), as is required to ultimately
calculate the jet transport ($T_{jet}$) [Beal et al., 2015].
To assess the accuracy of the transport proxy, the HYCOM transport proxy was compared
to the simulated (native) transport in HYCOM to quantify the differences between the
proxy and modelled transports and hence understand which processes the proxy may fail
to represent. The transport across the ACT section in HYCOM was extracted by setting
up the grid points between the two coordinates defining the start and end of the section
following the great circles of the sphere and calculating the defined transport at each
grid point along the section. The transport calculation facilitated a separation of the
transports into two components: the box transport ($T_{\mathrm{box}}$) and the jet transport ($T_{\mathrm{jet}}$).

### 2.4.2 Model SSH

In order to reproduce the "along-track" SSH altimeter data needed to create the proxy as
in Beal and Elipot [2016], 34 years of HYCOM SSH was linearly interpolated onto the
coordinates of the TOPEX/Jason satellite track number 96 overlapping the model ACT
array. The coordinates of the along-track altimeter data were obtained from the filtered 12
km Jason-2 Aviso satellite product, and not the unfiltered 6 km product which was used
for the original ACT proxy [Beal and Elipot, 2016], since the 12 km product matched the
~10 km model resolution more closely. To obtain the sea surface slope for each regression
model, an optimal pair of SSH data points was chosen such that the horizontal length



²⁷⁴ scale between them allowed for a maximum correlation between the sea surface slope

²⁷⁵ and *Tx*. The length scales of the slopes ranged from 24 km at mooring A to 12 km at

²⁷⁶ mooring G and 48 km for the offshore CPIES-pairs, indicating an increase in the spatial

²⁷⁷ scale of offshore flow, possibly due to increased offshore variability. Results from the *in*

²⁷⁸ *situ* proxy experiment by Beal and Elipot [2016] also showed an increasing length scale

²⁷⁹ with increasing distance offshore, however the results varied considerably in magnitude:

²⁸⁰ 27 km at mooring B to 102 km at mooring G. In this study the SSH slope was calculated

²⁸¹ such that a negative SSH slope corresponds to a negative surface velocity (southwest)

²⁸² according to geostrophy, whereas a positive slope would indicate positive northeastward

²⁸³ flow.

²⁸⁴ **2.4.3  Building the regression models**

²⁸⁵ Nine linear regression models were first developed to estimate the transport per unit

²⁸⁶ distance *(Tx* and *Txsw)* from the HYCOM sea surface slope during approximately the

²⁸⁷ same three-year period over which the ACT proxy was developed (April 2010- February

²⁸⁸ 2013). The three-year time period will further be referred to as the reference period.

²⁸⁹ Further tests were later performed, where the proxy was calculated over a range of different

²⁹⁰ time periods (see section 2.6).

²⁹¹ To calculate the total transport across the ACT array requires continuous *Tx* estimates

²⁹² across the current. This was achieved as in Beal and Elipot [2016] by fitting a piece-

²⁹³ wise cubic Hermite interpolating polynomial function to obtain transport estimates at 1

²⁹⁴ km intervals from the coast to the end of the array (Figure 2). Fitting the transport

²⁹⁵ function to the coast and equating it to zero would be equivalent to implementing a no

²⁹⁶ slip boundary condition in the model. Before calculating the total transport the current

²⁹⁷ boundaries needed to be defined. The box transport ($T_{\text{box}}$) was calculated by integrating

²⁹⁸ *Tx* horizontally to 230 km offshore, the three-year mean width of the current in HYCOM.

²⁹⁹ The jet transport ($T_{jet}$) was calculated using the algorithm developed by Beal et al., 2015

³⁰⁰ by integrating *Txsw,* the southwest transport component, to the first maximum of *Tx*

³⁰¹ beyond the half-width of the current (115 km in HYCOM) at each time step (Figure 2).

³⁰² Beal et al. [2015] argued that $T_{jet}$ therefore captured the southwestward transport of the




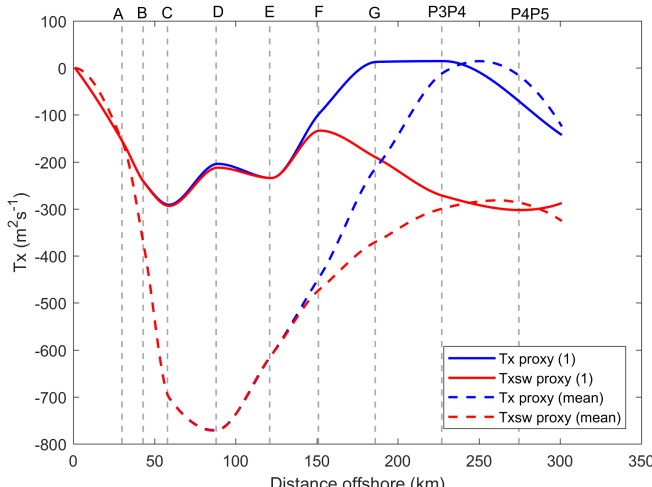

Figure 2: HYCOM transport per unit distance proxy (m$^2$ s$^{-1}$) for $Tx$ (blue) and $Txsw$ (red) transport at 1 km intervals at the first model time step (solid lines) and for the mean reference period (dashed lines). The faint grey lines represent the positions of moorings and offshore CPIES pairs.

meandering Agulhas Current. $Tx$ and $Txsw$ are simply shown at the first model time step
(week of the 3$^{\mathrm{rd}}$ of January 1980) in HYCOM and for the mean of the reference period
(2010-2013) to show the difference between the net and southwest transport components
used to calculate $T_{box}$ and $T_{jet}$ (Figure 2).
In order to test the accuracy of the transport proxy, it was first compared to the HYCOM
transport for the same period over which the proxy was developed (2010-2013). By
studying the corresponding model fields we were able to identify dynamic features in the
model that the proxy failed to capture. The correlation for the overlapping transports from
the model and the model proxy was calculated as well as the 3-year mean and standard
deviation (Table 3). Then, assuming that the three-year linear relationship between SSH
slope and transport per unit distance ($Tx$ and $Txsw$) from 2010-2013 remains constant,
the regression models were applied to the entire 34-year SSH model data. This resulted
in transport per unit distance estimates ($Tx$ and $Txsw$) for each mooring position at
each time step from 1980 to 2014. Thereafter, the 34-year transports were calculated by
applying the same methods that were used to calculate the 3-year transport time-series;
firstly, obtaining $Tx$ estimates at 1 km-intervals along the array and secondly integrating





horizontally to obtain $T_{box}$ and $T_{jet}$ (Figure 2).

## 2.5 Comparison of the transport proxy to actual model transports

The simulated model transports were calculated using the full-depth velocity fields across
the array. If the relationship between SSH slope and transport is strong, there would
be good agreement between the proxy and the actual model transports. To quantify
this correlations and transport statistics for the model and proxy were calculated from
the two-time series. These provided insight into which processes the proxy may have
failed to capture, which were then further investigated in HYCOM. Statistics are deemed
significant at the 95% significance level.
Eddy kinetic energy (EKE) was calculated to show the surface variability of the current
coincident with averaged SSH contours used to represent the mean surface structure
(Figure 5). The eddy kinetic energy was calculated as follows:

$$EKE = \frac{(u')^2 + (v')^2}{2} \tag{1}$$

where $u'$ and $v'$ are the zonal and meridional geostrophic current anomalies relative to the
geostrophic current mean calculated over the 3-year mean reference period, and over the
highest and lowest correlated years. In order to evaluate the subsurface current structure
along the ACT array, vertical velocity profiles were analysed for each mooring and CPIES-
pair over the 3-year mean reference period as well as over the highest and lowest correlated
years.
Transport variability in the HYCOM model was analysed by investigating residual trans-
port events in the worst and best performing regression models. In order to examine
the impacts of variable mesoscale features, residual transport events were identified as the
outlying residual transport values above and below 2 standard deviations of the estimated
transport.

$$e = Yi - \hat{Y}i \tag{2}$$

where $e$ is the estimated residuals, $Yi$ is the HYCOM transport per unit distance value
$(Tx)$ and $\hat{Y}i$ is the estimated transport per unit distance value according to the linear
regression models.



To investigate the current structure during these residual events, composite averages of
the cross-track velocity structure were analysed. The cross-track velocity at each depth
layer in HYCOM was extracted at 12 km intervals from 0 km to 400 km offshore, for the
34-year model period. Although the ACT array only reached 300 km offshore, analysis of
the current structure in HYCOM was extended further offshore. Previous analyses have
shown increased levels of offshore variability in this HYCOM simulation [Backeberg et al.,
2009; 2014], which therefore made it interesting to study the subsurface structure during
the offshore current meanders and the influence these could have on the transport proxy.
To further investigate the effect of the residual transport values on the box transport
proxy, considering it performed better than the jet transport proxy (see section 3.2), all
corresponding transport events exceeding plus or minus two standard deviations were
removed from each linear regression model during development of the proxy, after which
the $T_{box}$ proxy was re-calculated as explained in section 2.4.3 and evaluated against the
initial box transport proxy.
## 2.6   Sensitivity tests
To test the sensitivity of the time span of observations used to create the transport
proxy, sensitivity experiments were performed to test how many years of virtual *in situ*
observations are needed to create an accurate proxy to monitor the Agulhas Current
transport. Using 34 years of model data the linear relationship could be tested over much
longer or shorter periods.
Using the method described in section 2.4.3, regression models were built for 1, 6, 12, 18
and 34 years. In addition, the models were calculated over two arbitrary 3-year periods,
to test the influence that different current dynamics over different years could have on the
development of the transport proxy. Lastly, the regression models were calculated over
the maximum and minimum annual transport years in HYCOM, as well as during the
years the HYCOM transport standard deviation was the largest and the smallest. Table 2
shows the time range over which the sensitivity experiments were performed. The 3-year
*in situ* period in the model corresponded to the actual time range over which the *in situ*
experiment was conducted, April 2010- February 2013 [Beal et al., 2015].





Table 2: Sensitivity experiment time periods.

| Time range (years) | Model dates |
|---|---|
| 1 | Jan 2011 - Dec 2011 |
| 3 | Apr 2010 - Feb 2013 |
| 6 | Jan 2009 - Dec 2014 |
| 12 | Jan 2003 - Dec 2014 |
| 18 | Jan 1997 - Dec 2014 |
| 34 | Jan 1980 - Dec 2014 |
| 3* | Jan 1980 - Dec 1982; Jan 2000 - Dec 2002 |
| Max (Min) HYCOM transport. | 2003 (1982) |
| Max (Min) HYCOM transport STD. | 2013 (1980) |

3* Corresponds to the two additional 3-year periods

## 3 Results

### 3.1 HYCOM linear regression models

The coefficient of determination ($R^2$) from the regression models showed how well the linear relationship predicts the transport per unit distance estimates in HYCOM (Figure 3). The $R^2$ statistics from the regression models ranged from 0.86 at mooring A (30 km offshore) to 0.49 at the last CPIES-pair P4P5 (275 km offshore) for $Tx$ and 0.86 at mooring A to 0.37 at P4P5 for $Txsw$ (P values $< 10^{-3}$). Results from the *in situ* experiment showed an increase in the $R^2$ statistics in the regression models ranging from 0.51 at mooring A and 0.81 for CPIES-pair P4P5 for $Tx$ [Beal and Elipot, 2016], thus showing that the regression models had poorer skill inshore during the *in situ* experiment, whereas in HYCOM the regression models have poorer skill offshore. The results from the $Txsw$ regression models in HYCOM showed similar results for the inshore mooring locations (A, B, C, E) with slightly higher correlations for offshore moorings F, G and CPIES-pair P3P4 but a lower correlation for D and the furthest CPIES-pair P4P5. This shows that the $Txsw$ regression models explained more variance for moorings F, G and P3P4 but less variance for D and P4P5 than the $Tx$ regression models.





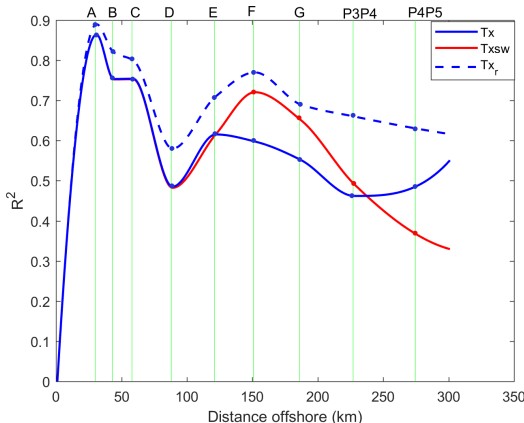

Figure 3: $R^2$ statistics from the linear regression models showing the relationship between HY-COM SSH slope and HYCOM transport per unit distance for each mooring (A-G) and CPIES-pair (P3P4 & P4P5) over the 3-year reference period (2010-2013). $Tx$ is represented by the solid blue line and $Txsw$ by the solid red line. The dashed blue line represents the results of $Tx$ after the removal of the residual transport events (see section 3.4). Sites A - CPIES pair P4P5 are shown by the faint green lines.

### 3.2 Proxy validation

In order to test the accuracy of the box and jet HYCOM transport proxies, these were compared to the box and jet transports extracted from HYCOM. This aided the investigation in terms of identifying transport events or features the proxy failed to represent.

Based on the correlation of the 3-year proxy transport (2010-2013) to the model transport over the same period, the box transport proxy explained 57% of the variance while the jet transport proxy only explained 14% of the variance. Assuming a constant three-year linear relationship for the nine regression models, the transport proxy was calculated using 34 years of HYCOM SSH slope, after which the 34-year box transport proxy explained 52% of the variance and the jet transport proxy explained 26% of the variance.

Table 3 summarises the transport statistics based on the 3-year and extended 34-year time period. The 34-year mean transport and standard deviation from HYCOM for the box and jet transport was -84 ± 47 Sv and -110 ± 38 Sv respectively. The proxy box transport was -87 ± 34 Sv and the jet transport was -92 ± 31 Sv. A higher jet transport was expected considering it excludes northeast counter-flows that decrease the box transport [Beal et al., 2015]. The differences between the standard deviations between HYCOM and



Table 3: a) Summary of the transport statistics of the HYCOM model transport against the HYCOM proxy transport over the 3-year and extended 34-year time period. Negative values denote transport in the southwest direction. 1 Sv=$10^6$ m$^3$s$^{-1}$. b) Correlations between the HYCOM model transport and HYCOM proxy transport, for the box transport and jet transport with the percentage of variance shown in brackets. All correlations were significant.

| a) | HYCOM (2010-2013) | | Proxy | | HYCOM (1980-2014) | | Proxy | |
|---|---|---|---|---|---|---|---|---|
| Transport | $T_{box}$ | $T_{jet}$ | $T_{box}$ | $T_{jet}$ | $T_{box}$ | $T_{jet}$ | $T_{box}$ | $T_{jet}$ |
| Mean & Std (Sv) | -81 ± 53 | -112 ± 41 | -91 ± 35 | -92 ± 30 | -84 ± 47 | -110 ± 38 | -87 ± 34 | -92 ± 32 |
| Max (Sv) | -223 | -244 | -196 | -185 | -236 | -245 | -213 | -219 |
| Min (Sv) | 44 | -48 | -36 | -46 | 87 | -30 | -20 | -27 |

| b) | $T_{box}$ | $T_{jet}$ |
|---|---|---|
| 2010-2013 | 0.75 (57%) | 0.38 (14%) |
| 1980-2014 | 0.72 (52%) | 0.51 (26%) |

the proxy indicate that transport in HYCOM experiences more variability compared to the proxy. The proxies only capture a portion of the transport estimate from the HYCOM model, suggesting it also only captures a portion of the model variability. The positive minimum transport values for $T_{box}$ during both time periods also appear to be peculiar, suggesting a current reversal during those events (Table 3).

Figure 4 shows the correlation between proxy and model transports for each year. The correlation per year for $T_{jet}$ varies greatly from year to year with a significant maximum correlation of 0.82 (2014) and an insignificant minimum correlation of 0.00 (2003). In contrast, the correlations for $T_{box}$ vary much less and are always significant with a maximum correlation of 0.88 (1988) and minimum correlation of 0.50 (1994). The box transport has higher correlations for most of the 34-year time period except during two single years where the jet transport has a higher correlation, 0.78 against 0.70 during 1991 and 0.54 against 0.50 during 1994. In summary, the results indicate that the proxy is generally better suited in HYCOM to estimate the box transport rather than the jet transport. Further analysis in this study therefore only focuses on the box transport.





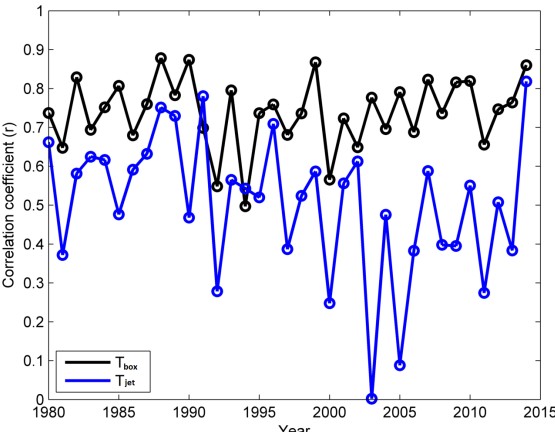

Figure 4: 34-year annual correlations between the box (black) and jet (blue) transport proxies against the box and jet transports extracted from HYCOM.

### 3.3  Evaluating the net transport proxy

The strengths and weaknesses of the box proxy are further investigated by selecting the highest and lowest correlated years from the 34-year annual correlations (Figure 4), and evaluated by plotting the current structure in the model over the respective years (Figures 5 & 6). Investigating the full-depth current structure could emphasize important subsurface processes which may not have distinct signatures at the surface and may therefore be excluded in the transport proxy.

Figure 5 shows the surface variability by displaying the eddy kinetic energy and the mean surface geostrophic flow as represented by the overlaying SSH contours over the 3-year reference period, and over the highest (1988) and lowest (1994) correlated years of the box transport proxy. During the reference period the current appears to be stable with low levels of EKE inshore whereas offshore the flow is more variable with higher levels of EKE. The flow depicts a similar structure during the lowest correlated year, however, during the highest correlated year the mean EKE is higher along and downstream of the array with a relatively stable current structure in comparison to 1994 and 2010-2013. The narrow spacing of the SSH contours for all three periods indicates a strong gradient inshore and hence a strong mean geostrophic current, however the wide spacing between the SSH contours offshore suggests that the variability in the model is confined to the offshore




side of the current. It is assumed that high levels of mesoscale variability in the model could bias the current position and hence the transport estimate, however, based on the analysis there were approximately five anticyclonic eddies during the highest correlated year (1988) and ∼7 anticyclonic eddies during the lowest correlated year which does not greatly differentiate the accuracy of the proxy for those years.

Figure 6 shows the mean cross-track velocity profiles during the reference period (2010-2013), the highest correlated year (1988) and the lowest correlated year (1994) for each mooring and the CPIES-pairs. The model cross-track velocity changes direction with depth, specifically for offshore mooring G and CPIES-pairs P3P4 and P4P5, at the depth of ∼2000 m (Figure 6) thereby defining the depth of the Agulhas jet. During the 3-year reference period the velocity changes direction at moorings B and G (∼1200 m and ∼2000 m respectively) and at sites P3P4 (∼2000 m) and P4P5 (∼300 m, ∼2000 m). During 1988 sites F-P4P5 experience a change in direction (>∼2000 m). Lastly, during 1994 mooring G and sites P3P4 and P4P5 exhibit a change in direction (>∼2000m). This shows that the offshore variability in the model impacts not only the surface variability (Figure 5) but also the subsurface flow, which would directly impact the accuracy of the box transport proxy.





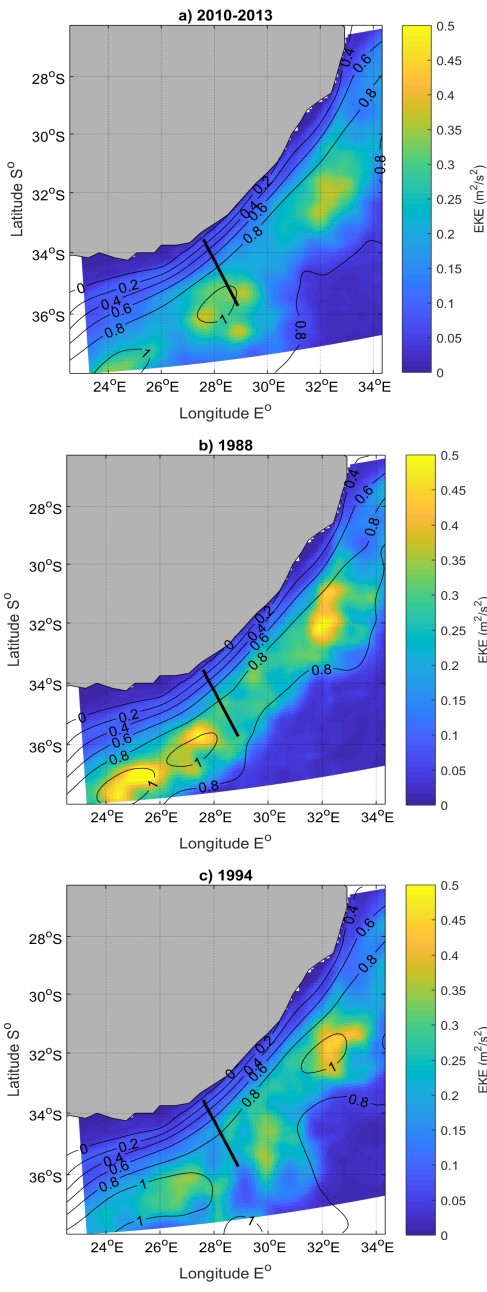

Figure 5: Eddy kinetic energy (EKE in $m^2s^{-2}$) and sea surface height (SSH in m) contours during (a) the reference period (2010-2013) (b) the highest (1988) and (c) lowest (1994) correlated years. The black line representing the ACT array.





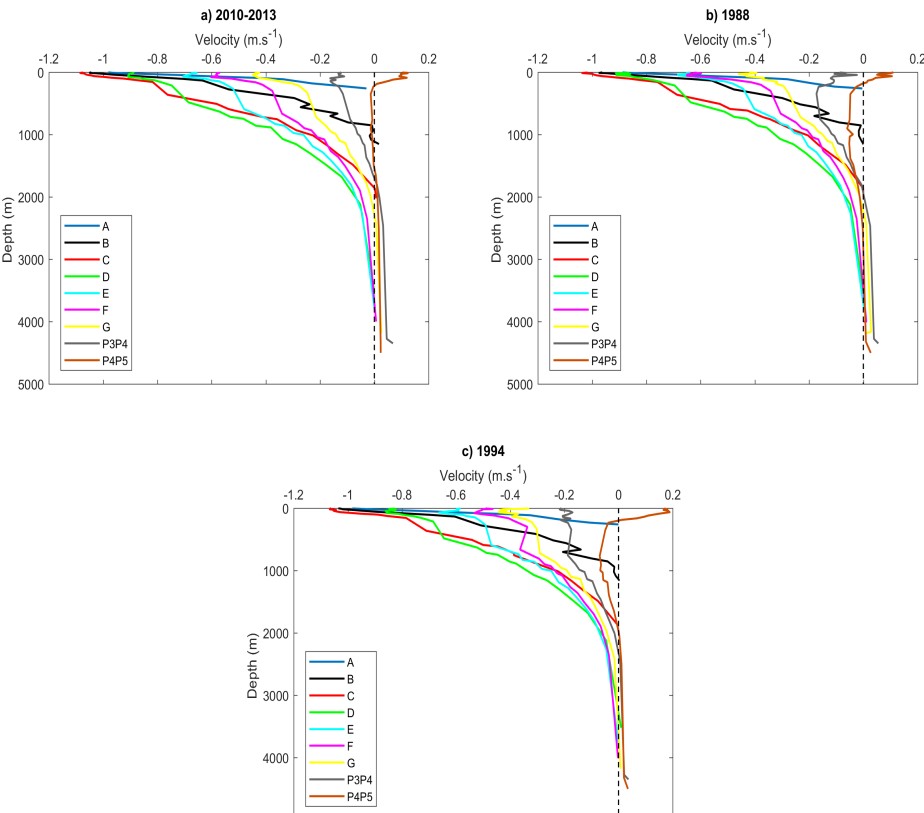

Figure 6: Mean cross-track velocity profiles (m s$^{-1}$) during (a) the 3-year reference period (2010-2013), (b) during the highest correlated year (1988) and (c) the lowest correlated year (1994). Each colour represents the different moorings (A-G) and CPIES-pairs (P3P4 & P4P5) . Negative values indicate southwestward flow.

### 3.4   Investigating the transport variability

This section will investigate factors of transport variability in the HYCOM model which caused the limitations in the HYCOM transport proxy. It was previously shown that the performance of the linear regression models weakened moving offshore because of the decrease in correlation between transport per unit distance and SSH slope. Regression model 8, CPIES-pair P3P4 (RM 8, Figure 7a), captured the least transport variance at 46% and regression model 1, mooring A (RM 1, Figure 7b), explained the most transport variance at 86%. The differences between the magnitudes of the residual transport events between RM 1 and RM 8 emphasize a large difference in transport variability between



the inshore and offshore mooring locations in HYCOM.
According to the methods presented above, a negative SSH slope in HYCOM corresponds
to a negative (southwest) surface velocity and if the current structure were barotropic, a
negative (southwest) transport per unit distance estimate and vice versa. As shown in
regression model 1 (Figure 7b), all the data points are clustered such that the negative
SSH slope relates to a negative transport per unit distance, in the absence of northeast
counterflows. Careful analyses of regression model 8 shows that eight of the nine resid-
ual transport events do violate the proportional relationship between SSH slope and Tx
(Figure 7a). Some of which have a negative SSH slope relating to a positive transport per
unit distance where others show a positive SSH slope with negative transport per unit
distance. Therefore the SSH slope does not always reflect the direction of flow at depth,
and thus the correct sign for $Tx$.
Examination of the cross-track velocity structure with depth (Figure 8) shows that there
is a change in the direction of velocity in the bottom layers at the location of regression
model 8 (CPIES-pair P3P4). The cross-track flow in the surface layers ($\sim$0-700 m) of the

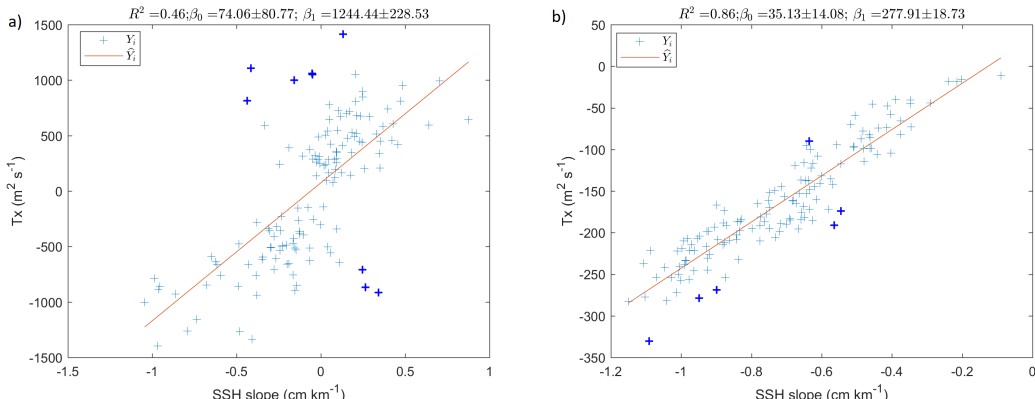

Figure 7: Linear regression models showing the relationship between HYCOM SSH and transport per unit distance ($Tx$) for a) CPIES-pair P3P4 (RM 8); capturing the least transport variance (46%) and b) Mooring A (RM 1); capturing the most transport variance (86%). $Yi$ (blue crosses) represent the Tx values from HYCOM and $\hat{Y}i$ (red line) represents the Tx estimates from the linear regression model. The bold crosses highlight the residual transport events with transport values greater or less than 2 standard deviations of the transport estimate. The coefficient of determination ($R^2$) quantifies the amount of variance explained by the regression model, $\beta\iota$ is the slope coefficient and $\beta o$ the intercept with 95% confidence intervals. Note the different scaling on the x & y-axes.





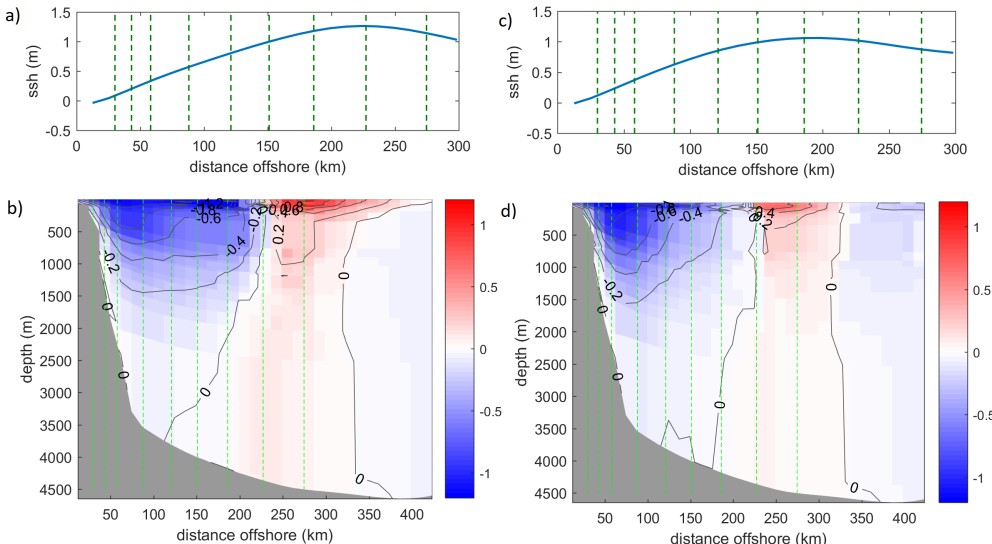

Figure 8: Composite SSH (m) and cross-track velocity structure ($ms^{-1}$) of the residual transport events from a & b) regression model 8 and c & d) regression model 1. Blue shading represents the negative, southwest current direction and red represents the positive, northeast current flow. Contours are every $0.2\ ms^{-1}$. Dashed vertical lines represents the nine locations of the mooring and CPIES-pairs, the first line representing mooring A and CPIES-pair P4P5 furthest offshore.

current is towards southwest, whereas below ∼700 m the flow is towards the northeast.
Therefore, the vertically integrated flow (Tx) is positive, that is towards the northeast,
and in the opposite direction implied by the SSH slope. In contrast, at the location of
mooring A, the composite velocity field is always towards the southwest, that is consistent
with the SSH slope.
The residual investigation (Figures 7 & 8) shows how large outliers decrease the overall
performance of the linear regression models, by decreasing the percentage of captured
variance. If these transport events were removed the performance of the linear regres-
sion models would statistically increase. Removing the outliers larger than ±2 standard
deviations from regression model 8, increases the percentage of captured variance from
46% to 66%. For model 1, removing outliers increases the captured variance from 86% to
88%. The improvement is specifically greater for regression model 8 due to the removal of
the extreme events that violated the directly proportional relationship between SSH slope



and transport. Figure 3 shows the increase in the performance of the linear regression
models after the removal of the outlying transport events from all nine regression models.
The increase in variance explained is notable for the regression models corresponding to
the inshore moorings B and C and offshore moorings F, G and CPIES-pairs P3P4 and
P4P5. After the removal of the outlying transport events, the box transport proxy was
re-calculated and its performance compared to the initial proxy. The "improved" $T_{box}$
proxy captures more variance, 72%, compared to 52% for the original proxy.

## 3.5 Sensitivity tests

The 34-year Agulhas transport proxy was based on regression models built using only
3 years of HYCOM model data. The statistics in Table 4 and Figure 9 illustrates the
results obtained from building the linear regression models and deriving the transport
proxy using 1, 3, 6, 12, 18 and 34 years of model data. The Taylor diagram (Figure
9) shows the distribution of the results in terms of standard deviation of the transport,
the correlation, and the root-mean-squared error (RMSE) between the proxies and the
HYCOM model transport. We find that the correlation between proxy box transport
and model box transport is not improved by using more model data to build the proxy.
The correlation is 0.72 when using data from 2010-2013, and changes by no more than
0.01 when extending the number of years of model data. Similarly, building the proxy
with one year of model data decreases the correlation by only 0.01 (Figure 9 & Table 4).
The only visible difference was the decrease in standard deviation. It was expected that
the correlation would increase because using more years of model data may capture more
current variability and the RMSE would decrease to correspond to the model transport
estimates.
The sensitivity of the box transport proxy was also tested using two arbitrary 3-year peri-
ods. In comparison to the correlation obtained during 2010-2013 the correlation decreased
by 0.02 during 1980-1982 and remained the same during 2000-2002. The results obtained
from calculating the $T_{box}$ proxy during the maximum (minimum) transport and standard
deviation years in HYCOM showed no improvement or decrease in the skill of the proxy
either.




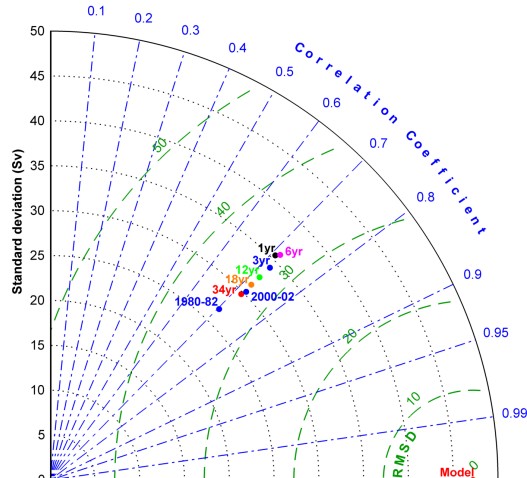

Figure 9: Taylor diagram showing the results of the box transport proxy calculated based on a 1-year linear relationship (black), 3-years (blue), 6-years (magenta), 12-years (green), 18-years (orange), 34-years (red) and during 1980-1982 and 2000-2002 (blue).

Table 4: Transport statistics and correlation results obtained from calculating the net transport proxy over a range of time periods.

| Net transport | Transport (Sv) | STD (Sv) | RMSE (Sv) | r |
|---|---|---|---|---|
| **MODEL** | -84.32 | 47.23 | 0 | 1.00 |
| **1-yr** | -87.26 | 35.47 | 33.36 | 0.71 |
| **3-yr** | -87.21 | 34.09 | 32.76 | 0.72 |
| **6-yr** | -87.04 | 35.91 | 33.04 | 0.72 |
| **12-yr** | -86.91 | 32.51 | 32.83 | 0.72 |
| **18-yr** | -88.71 | 31.28 | 32.95 | 0.72 |
| **34-yr** | -88.15 | 29.74 | 33.14 | 0.72 |
| **1980-1982** | -87.86 | 26.80 | 34.14 | 0.70 |
| **2000-2002** | -94.80 | 30.31 | 32.87 | 0.72 |

## 4 Summary and conclusions

The Agulhas Current transport proxies, developed by Beal and Elipot [2016], were based on nine linear regression models, each assuming a constant linear relationship from three years of observations between *in situ* transport and satellite along-track sea surface gradients. Applying constant linear models and assuming a constant vertical current structure, the transport proxies were extended using 22-years of along-track satellite data in order to yield two 22-year time-series of Agulhas Current transports [Beal and Elipot, 2016]. The Agulhas Current transport proxies in the current study replicates the methods used by





Beal and Elipot [2016] but applies these using a regional HYCOM model of the Agulhas
Current [Backeberg et al., 2009; 2014]. The HYCOM transport proxies were developed
using nine, three-year linear regression models between model transport and model SSH
slope, and extended using 34-years of the model SSH data from 1980 to 2014.
The HYCOM model provided the means to investigate the validity of the assumptions used
to create the proxies, such as the constant relationship between SSH slope and transport
per unit distance at each mooring location and the temporal scale of observations needed
to build a strong linear relationship between transport and SSH slope. Two transport
types, the box transport and the jet transport, were extracted from HYCOM in order
to validate the box transport proxy ($T_{\text{box}}$) and the jet transport proxy ($T_{\text{jet}}$). The $T_{box}$
proxy explained a higher percentage of transport variance (57%) during the three-year
reference period (2010-2013), in comparison to the $T_{jet}$ proxy that only captured 14%
of the variance. Using 34-years of model data (1980-2014), assuming the fixed 3-year
relationship between SSH slope and transport, $T_{box}$ explained 52% of the variance in
comparison to $T_{jet}$ that only captured 26%. Results from Beal and Elipot [2016] also
showed that the box transport proxy ($T_{box}$) explained a higher percentage of variance
(61%) during the ACT period than the jet transport proxy ($T_{jet}$: 55%).
The poorer performance of the $T_{jet}$ proxy in HYCOM compared to the *in situ* $T_{jet}$ proxy
of Beal and Elipot [2016] is partly due to various model discrepancies such as the consist-
ent merging of the anticyclonic eddies with the Agulhas Current in the northern region
[Backeberg et al., 2014], in addition to unresolved eddy dissipation in this region [Braby
et al., 2016]. It may also possibly be because it only represents the southwestward flow,
whereas the input sea surface slope reflects the net flow along the array. Therefore, con-
sidering the box transport proxy explains a higher percentage of variance for most of the
34-year period, further analysis on the current structure was based on the $T_{box}$ proxy only.
One of the main assumptions on which the Agulhas transport proxy relies is that the
vertical structure of the current does not change outside the 3-year reference period [Beal
and Elipot, 2016]. There are limitations to the ability of satellite altimeters to detect
sub-surface variability [Robinson, 2004], however, it has been suggested that a strong
relationship between SSH and full-depth transport exists [Beal and Elipot, 2016].




The surface structure of the current was investigated in terms of the mean EKE and SSH
contours (Figure 5), which are ideally equivalent to surface geostrophic flow and hence
show the mean horizontal extent of the current [Robinson, 2004]. The vertical variability
was investigated by plotting the mean cross-track velocity profiles (Figure 6). During the
highest correlated year (1988) the current is stable and inshore, whereas during the lowest
correlated year (1994) and during the proxy development period (2010-2013) the current
is meandering and it appears that a large portion of the energy of the current has been
shifted offshore. These results are consistent with Elipot and Beal [2015], who showed that
during the passage of a meander event, a large portion of kinetic energy is extracted from
the flow through the process of barotropic conversion. Results from the analysis of the
vertical profile of the current reveals subsurface counterflows, specifically for the offshore
moorings (G, P3P4 and P4P5) and occasionally for inshore mooring B. An explanation
for the offshore subsurface counter flows may be due to the impinging baroclinic eddies
continuously propagating downstream [Backeberg et al., 2009], which thereby affect the
entire water column by changing the direction of flow at certain depths. This will explain
why the transport proxy fails to capture current reversals, as implied by the positive
minimum transport values in Table 3, because the SSH slope is not reflective of the
subsurface counterflows associated with the impinging baroclinic eddies. The occasional
current reversal for inshore mooring B (43 km offshore, 1264 m depth) may be due to
influence of the simulated Agulhas Undercurrent in HYCOM which flows approximately
40-60 km offshore, 1000-1700 m deep (Figure 8), as opposed to *in situ* estimates of 11-60
km offshore and 1000-2900 m deep [Beal, 2009].
The question still remains as to why most of the transport variance was explained in the
year 1988 and the least in 1994? Figure 3 highlighted that the performance of the linear
regression models decreased offshore, such that when the current is in a meandering
state, the $T_{box}$ proxy fails to accurately estimate the transport. It could be assumed
that using the $T_{jet}$ proxy would improve the accuracy, however, the performance of the
southwest regression models are only slightly stronger at the offshore end of the array. The
jet transport proxy by Beal and Elipot [2016] was developed to effectively estimate the
transport of the Agulhas Current in the event of a mesoscale meander, which generally
causes the current to manifest as a full-depth, surface intensified, cyclonic circulation



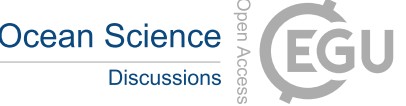

out to 150km from the coast with anticyclonic circulation farther offshore [Elipot and
Beal, 2015]. The Agulhas meanders in the HYCOM simulation occur in association with
large anticyclonic eddies predominantly defined to the offshore edge of the current, with a
narrow, southwest stream against the coast [Backeberg et al., 2009] or in some instances
with an anticyclonic eddy across the entire length of the array. The resolution of HYCOM
is able to capture the mesocale dynamics of eddies [Holton et al., 2017] however, it fails to
resolve the near-coastal features, such as the inshore, surface intensified cyclonic motion
in this simulation. This would require a finer resolution at the coast, in order to reveal
smaller offshore displacements, ∼50 km, associated with these meander events [Elipot
and Beal, 2015]. The high levels of offshore variability in HYCOM is therefore the main
limiting factor in the performance of both transport proxies.
The regression model for CPIES-pair P3P4 (regression model 8) performed the worst, only
explaining 46% of the transport variance (Figure 7a). Evidence from the HYCOM velocity
fields showed that the offshore location of CPIES-pair P3P4 was highly susceptible to the
impinging anticyclonic eddies, which in turn resulted in high levels of variability in the
horizontal and vertical velocity current structure (Figures 7a & 8). The presence of the
anticyclonic eddies would be included in $T_{box}$, considering that the eddies produce a strong
surface signature (Figure 5), but the SSH slope might not necessarily be reflective of the
transport beneath the eddy. It has been observed in a layered ocean that, when assuming
geostrophy, the net transport in the uppermost layer (∼0-1000 m) is mainly proportional
to the SSH slope [Andres et al., 2008]. If this was the case, the performance of regression
model 8 would be higher, but the current experiences a baroclinic flow beneath the entire
water column which is not reflective of the SSH slope (Figure 8). As the anticyclonic eddy
crosses the offshore edge of the ACT array, its baroclinic nature in HYCOM effects the
direction of velocity beneath the location of CPIES pair P3P4, which therefore results in
a weak correlation between SSH slope and transport. The impinging anticyclonic eddies
would have a similar influence on the offshore regression models for mooring G and CPIES
pair P4P5 (Figure 8).
The regression model for mooring A (regression model 1) performed the best in terms of
the correlation between SSH slope and transport per unit distance (Figure 7b) explaining
86% of the variance. The inshore location of mooring A, 30 km off the coast, experienced



low levels of transport variability with a stable southwest current trajectory (Figure 5 &
8) and Figures 6 and 8 illustrate a barotropic current structure in the vicinity of mooring
A with no sub-surface counterflows. The small, sub-surface variability observed inshore of
the array, below 2000 m depth do not necessarily have a direct impact on the SSH signal or
drastically change the volume transport of the water column, however the further offshore,
the closer the interaction of the current with the offshore baroclinic eddies, the weaker
the performance of the regression model.
It is important to consider that the Agulhas Current simulation in HYCOM is not com-
pletely realistic, demonstrating much higher levels of mesoscale variability than observed
[Backeberg et al., 2008; 2009]. Rouault and Penven [2011] and Elipot and Beal [2015]
showed that, on average, 1.6 mesoscale meanders pass through the ACT array at 34°S per
year. In the HYCOM simulation an average of 5 anticyclonic eddies passed over the array
per year. A study by Braby et al. [2016] investigating eddy activity in the northern Agul-
has Current using satellite altimetry, showed that both cyclonic and anticyclonic source
eddies dissipate upon approaching the main Agulhas Current. However, the observed
eddy interaction and dissipation process is poorly resolved in many numerical ocean mod-
els [Tsugawa and Hasumi, 2010; Penven et al., 2011; Durgadoo et al., 2013; Backeberg
et al., 2014; Loveday et al., 2014], including the HYCOM model used in this study.
The frequently impinging eddies make it difficult to effectively estimate the accurate box
transport of the Agulhas Current in the model since the advection of these eddies have
previously been found to be responsible for large transport fluctuations [Backeberg et al.,
2009]. The transport proxy only includes the transport of the portion of the eddy that
is reflected in the SSH signal across the array, whether it is only the southwestward or
northeastward portion of the eddy or both, and should therefore match the transport
peaks from the model. The transport in the model and proxy may fluctuate accordingly,
however the transport estimates will not necessarily be equivalent, since it also depends
on the strength of the proxy along the ACT array. In other words, the transport proxy
may capture the SSH signal of the eddies along the array, however the correlation of the
regression models decrease offshore, therefore transport estimates inshore would be more
accurate than the transport estimates offshore when the current is in a meandering state.
It was shown that removing the residual transport events violating the proportional rela-



tionship between SSH slope and transport improved the proxy performance i.e. increased
the percentage of transport variance explained. Several studies have researched methods
to decrease the levels of EKE in numerical simulations. Backeberg et al. [2009] improved
the representation of the southern Agulhas Current by applying a higher-order momentum
advection scheme, resulting in a well-defined meandering current rather than a continu-
ous stream of eddies. Anderson et al. [2011] found that the use of relative wind forcing
significantly decreased eddy intensities and a study by Renault et al. [2017] focussing on
the current stress feedback between the ocean and atmosphere demonstrated a reduction
of mesoscale activity by deflecting energy from the geostrophic current to the atmosphere,
showing that the indirect current feedback, improved the representation of the Agulhas
Current. Improving the mesoscale variability in the HYCOM model could therefore yield
better results for the transport proxy, specifically for the offshore regression models, in the
future. Furthermore, improving the simulation of coastal, shelf and continental slope fea-
tures, including the Agulhas Undercurrent could decrease the performance of the inshore
regression models. In order to effectively mirror the performance of the *in situ* transport
proxy developed by [Beal and Elipot, 2016] would ideally require a numerical model that
accurately simulates Agulhas meanders and the vertical variability, including an accurate
representation of the Agulhas Undercurrent, which has not yet been achieved in existing
regional configurations.
The development of the ACT transport proxy was initially tested using a regional NEMO
configuration in order to evaluate the potential of the altimeter proxy to monitor the
multi-decadal transport of the Agulhas Current [van Sebille et al., 2010]. Using the
numerical model, it was concluded that the correlation between the Agulhas Current
transport and gradient in sea surface height was greater than r=0.78 for any three-year
measuring period, and is therefore an adequate timescale to build an accurate transport
proxy [van Sebille et al., 2010].
The HYCOM output in the current study was used to test the validity of the relationship
between transport and SSH slope over a range of time periods. It was hypothesised
that building the linear relationship over longer time periods, >3 years, would increase
the skill of the transport proxy, since the linear relationship would include more current
variability over longer periods of time. The results showed that calculating the transport





proxy over longer or shorter time periods did not necessarily improve the performance of the proxy, thereby suggesting that the current dynamics for any 3-year period in the model could be very similar, in agreement with the results obtained in van Sebille et al. [2010], suggesting that the results were consistent despite the model biases. This justifies that 3-years is a sufficient time-period to develop the satellite-altimeter transport proxy of the Agulhas Current in HYCOM. Lastly, the study showed that the transport proxy is sensitive to subsurface variability in the model, suggesting that caution should be taken regarding the implicit assumption of a fixed vertical current structure. The accuracy of the transport proxy remains sensitive to model bias and implications therein, suggesting that these results should be tested rigorously in other model simulations. Sensitivity studies of this kind, using numerical ocean models, provide useful information into planning *in situ* studies in the future, and understanding the sensitivities and limitations of transport proxies could further improve long-term monitoring methods in the global ocean.





*Authors contributions*
E.V. conducted the data analyses and wrote up the final paper. B.B provided the HYCOM
model data, supervised the project and provided financial support. J.H. supervised the
project and provided financial support and S.E. assisted with the methodology of the
transport proxy. All authors helped to conceptualize ideas and contributed to writing the
paper.
We have no conflicts of interest to disclose.
*Acknowledgements*
This work has been funded by the National Research Foundation of South Africa and
by the bilateral South Africa-Norway SANCOOP SCAMPI project. We would like to
thank the Nansen-Tutu Centre in South Africa and SAEON for providing opportunities
to present the project locally and internationally. We thank the Nansen Environmental
Remote Sensing Centre (NERSC) in Bergen, Norway, for hosting us for a duration of
the project and wish to thank Dr. Knut-Arild Lisæter for his guidance while working at
NERSC. We gratefully acknowledge Professor Lisa Beal, Dr. Shane Elipot and the rest of
the ASCA team from the Rosenstiel School of Marine and Atmospheric Science (RSMAS),
University of Miami, for granting us permission to replicate the Agulhas transport proxy
methods. Shane Elipot was supported by the U.S. National Science Foundation through
the ASCA project, Award OCE-1459543.



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
