# Peer review of "Investigating the relationship between volume transport and sea surface height in a numerical ocean model"

_Ocean Science, 2018_

## Author Comment (AC1) · 3 Dec 2018

The Agulhas Current System plays a vital role in global-climate circulation, being the strongest western boundary current in the Southern Hemisphere. Several climate models have proposed that western boundary currents, such as the Agulhas Current, are becoming stronger due to the intensifying global wind systems and anthropogenic climate change. To validate such model predictions requires accurate long-term observational evidence. There is evidently a trade-off between spatial and temporal monitoring. In situ observations may accurately measure the dynamics of the Agulhas Current throughout the water column but are costly and spatially coarse. Whereas, satellite

observations can provide high-temporal and spatial data of the surface ocean but lacks detailed information below the surface. Numerous studies aiming to monitor long-term changes in global current systems have adopted methods to combine various sampling tools, including the development of the Agulhas transport proxy.

The Agulhas transport proxy developed by Beal and Elipot (2016) was built based on the physical principle of geostrophy, where along-track sea surface slope measured by an altimeter can be interpreted as a measure of the volume transport across a portion of the current, assuming the current depicts an equivalent barotropic structure with depth. This modelling study aimed to recreate the Agulhas transport proxy within a regional HYCOM of the Agulhas Current System, attempting to test the validity of the underlying assumptions on which the satellite-altimeter proxy was based. The 34-year regional-hindcast simulation from HYCOM provided the tools to test the sensitivity of the transport proxy to vertical changes in the current and the length scale of observations used to build a strong, constant linear relationship between volume transport and SSH slope.

This study motivates the need to improve long-term monitoring methods, where such improvements include advances in model development, combined with adequate validation studies, to help plan future experiments intending to monitor long-term changes in ocean circulation.

---

## Referee Comment (RC1) · Anonymous Referee #1 · 21 Dec 2018

In this manuscript, the authors use the HYCOM model to assess the accuracy of the ACT array, an observational array that has been in the water to measure the strength of the Agulhas Current and relate that to altimetry data, in order to create a long proxy record of Agulhas Current variability. The authors find that, in their HYCOM model, this approach to create a proxy is indeed feasible

This is an interesting and very thorough study, that could in principle be published in Ocean Sciences. However, before I can recommend publication, there are a few major issues that I suggest need to be resolved

1) The manuscript is very long, and would benefit much from a considerable shorten-

ing. Especially because of the structure with methods and results separate, there is a lot of duplication between the latter subsections of section 2 and the whole of section 3. Since the authors then repeat the results (again!) in section 4, this really makes for a tough read.

2) It is unclear what the immediate relevance of the study is. The authors present their results very much as a proof-of-concept for the ACT measuring principle, but the moorings have already been successfully deployed. The motivation therefore feels a bit redundant. Another motivation could be to improve physical understanding of the relationship between SSH and transport, but for that the manuscript is too much focussed on the statistics of the relation between the two variables, rather than the hydrodynamics. For example, there are quite a few statements (e.g. line 277 & 279) where a careful analysis of the hydrodynamics would be appropriate

3) The construction of Tjet and Tbox is quite confusing. For example, there is a Tx and a Txsw, even though in both cases they are used for the transport in the souwthwest (sw) direction? Use better terms for these? Might it help to add the equations how all these transport variables are constructed?

4) There is no validation of the depth structure of the Agulhas Current in HYCOM. Given that there is quite some mention of the baroclinic nature of the current, this would be good to validate using e.g. the ACT array data themselves

5) It is a missed opportunity I feel, that the authors have not also investigated the temperature/heat transport. That is something that was hard to do in the ACT array itself, yet is crucial for its climate monitoring ambition. Here, the authors have all the information to calculate the relation between volume and temperature transports

Other, more minors comments are

- The abstract is fairly technical and detailed, especially in the second half. I am not sure how relevant this is to most readers. For example, how useful is it to mention the

terms Tjet and Tbox if they are not explained?

- line 110: add 'time' before 'length scale'?

- line 161: It is unclear whether the nesting is one-way or two-way

- Is table 1 really relevant? Most, if not all, of the information is also in the text. And since there is only one model setup, why does it need to be in a table?

- Figure 1: The altimetry line stops just before reaching the shore. Is this an artefact of the plotting, or does this highlight that nearshore altimetry is not used. If the latter, it would be good to mention that

- line 272: I don't understand why the 12km product is used, if the 6km product is more accurate. Why not interpolate the 6km product to the actual mooring locations?

- Eq 2: Why not use Tx here, if it is equivalent to Yi?

- Table 3 would be much more useful if it also listed the observational ACT results?

- line 490: Is this increase from 86% to 88% is statistically significant?

- Table 4: I don't understand why all the r-values are essentially the same. What does this tell us about the system? How to interpret this? And how is the correlation with the observations?

type-os etc: - line 62: 'area' instead of 'field'?

- line 120: Zhu et al should be \citep{}

- line127: 'but may also be'

- line 182: remove 'notably'

- Figure 2: use 'dashed' instead of 'faint'?

- line 641: 'has' instead of 'have'

---

## Referee Comment (RC2) · Anonymous Referee #2 · 16 Jan 2019

This manuscript uses a numerical model of the western Indian Ocean, Agulhas Current region, to investigate the performance of the Beal and Elipot (2016) transport proxies that were developed using in situ mooring observations and satellite altimetry. While the intention of the study is sound, the study does not actually use the numerical model to full advantage. The investigation of the robustness of the proxy is limited and the study does not fully explore the dynamical reason why the proxy fails in some instances ("box") and almost entirely for the "jet". If the study was more thorough in identifying the dynamical reasons for failure of the proxy this study could be used as a basis for either suggesting and exploring a more appropriate region where a simple proxy as proposed may be of value or identify dynamical instances where the proxy is accurate.

[Figure]

Thus this study is potentially interesting, but fails to make use of the numerical model to fully explore the validity or not and why of the proposed Agulhas current proxy.

In addition to a more detailed dynamical exploration, the manuscript would benefit with a reorder and editing. For example, in the Summary and Conclusion section the authors state (line 534-537) "The HYCOM model provided the means to investigate the validity of the assumptions used to create the proxies, such as the constant relationship between SSH slope and transport per unit distance at each mooring location and the temporal scale of observations needed to build a strong linear relationship between transport and SSH slope." They then follow with a limited discussion explaining some reasons why the proxy does not capture the model transport, referring to figures to justify this reasoning – this is not a summary or conclusion. It is suggested that much of the information (lines 534-628) should be incorporated into the relevant parts of Section 3.

A reordering of Lines 629-696 would form what may be considered a "Summary and Conclusion" sections.

Section 2.1 should only provide details of the model used in this study. The reader is not interested in the details of the larger regional model that provided the boundary conditions of the higher resolution (1/10o) nested model.

The presentation of section 2 was convoluted and thus difficult for the reader to easily understand the approach taken. It is suggested that the authors revise this section to more clearly and concisely explaining the methods and assumptions.

Lines 275-290 "The length scales of the slopes ranged from 24 km at mooring A to 12 km at mooring G and 48 km for the offshore CPIES-pairs, indicating an increase in the spatial scale of offshore flow, possibly due to increased offshore variability. Results from the in situ proxy experiment by Beal and Elipot [2016] also showed an increasing length scale with increasing distance offshore, however the results varied considerably in magnitude: 27 km at mooring B to 102 km at mooring G." Can you explain the

reason for the difference in length scales between the model and observations (in situ and satellite)? Does this indicate the model doesn't capture the observed variability? What implications does this have for this study?

It is suggested that section 2.4.1 be revised to remove any unnecessary information concerning the larger regional model.

Line 407-408. "The proxies only capture a portion of the transport estimate from the HYCOM model, suggesting it also only captures a portion of the model variability." Is this the only problem with the proxy estimate? A more detailed analysis is really required to understand the impact of the assumptions used in developing the proxy.

Line 418-420 "In summary, the results indicate that the proxy is generally better suited in HYCOM to estimate the box transport rather than the jet transport. Further analysis in this study therefore only focuses on the box transport." It is not appropriate to simply ignore results that don't agree. You need to fully explore the reasons why the different proxies fail.

Lines 485-499 Removing outlier to increase the performance of the proxy is not appropriate. The authors should clearly identify the dynamical reasons for the reduced skill of the proxy. It is only through this in-depth analysis that advantages and disadvantages of the proxy can be fully explored.

The manuscript is lengthy and the prose overly convoluted and repetitive, when reviewing the manuscript the authors should, where possible, simplify the writing and remove repetition. Below are a few examples:

Line 85-89 "The Agulhas transport proxy of Beal and Elipot [2016] was derived from the physical principle of geostrophy, where along-track sea surface height slope measured by satellite altimeters can ultimately be related to a measure of volume transport across a portion of the current, provided that the surface current represents the flow at depth [Beal and Elipot, 2016]. " can be deleted as lines 89-93 fully explain the major findings

of the Beal and Elipot, 2016 study.

Line 151 change "... in doing so..." to ".. thus ..."

Line159-161 remove " The horizontal resolution of the parent model ranged from 14 km in the northern Indian Ocean to 45 km in the Southern Ocean, with a resolution ranging from 30 to 40 km in the region of the Agulhas Current." This information is not needed; the reader can refer to George et al., 2010 if they require more information on the model from which the boundary conditions were taken.

Line 154-155 Change "The HYCOM output in this study was made available from a nested 1/10° model of the greater Agulhas Current System (AGULHAS) [Backeberg et al., 2008; 2009; 2014]." To "This study used output from a nested 1/10° model of the greater Agulhas Current System (AGULHAS) [Backeberg et al., 2008; 2009; 2014]."

These are a few examples; there are many more instances of repetition and where more concise writing would improve the text.

Minor comments

Line 45 change "As the current continues southwestward the current becomes.." to "As the current continues southwestward it becomes.. "

Line 60-62 poorly constructed sentence "The unique circulation of the Agulhas Current System, in the context of regional and global climates, makes it an important field of research."

Line 67: "However, the close proximity of the current to the coast makes it difficult to monitor using satellite altimetry [Rouault et al., 2010]." Is this statement still true given the development of the AVISO X-track product (https://www.aviso.altimetry.fr/en/data/products/sea-surface-height-products/regional/x-track-sla/coastal-along-track-sea-level-anomalies.html)?

Line 74-84. It can be shown that the total cost of in situ observing, satellite observations

and models are all on similar cost. Singling out in situ observations as the only costly tool is not appropriate or accurate.

Change " In situ observations may accurately measure the dynamics of the Agulhas Current throughout the water column but are expensive and spatially coarse." To "In situ mooring observations provide high temporal observations of the Agulhas Current throughout the water column but spatially coarse."

Line 106 Change [Beal and Elipot, 2016] to Beal and Elipot [2016]

Line 120 Change Zhu et al. [2004] to [Zhu et al., 2004]

Line 158-159 Change ".. buffer zone." To ".. sponge layer."

Line 166-167 Change "Both models have 30 hybrid layers and targeted densities ranging from 23.6 to 27.6 kg/m3. To "AGULHAS has 30 hybrid layers and targeted densities ranging from 23.6 to 27.6 kg/m3."

Line 185 Add " . . . 2010-2013 (Figure 1, Beal et al., 2015).

Line 193-195 Change "During the first phase of the ACT experiment, the mooring array was maintained in the Agulhas Current for a period of 34 months, perpendicular to the continental slope at 34°S, south of East London, South Africa (Figure 1)." To " The ACT mooring array was located perpendicular to the continental slope at 34°S, south of East London, South Africa (Figure 1)."

Line 200 Change " From the data collected in Beal et al. [2015], two volume transports were estimated:. . . " to "From the data collected, Beal et al. [2015], provided two volume transports estimates: .."

Line 202 Change ". . . is a net transport" to ". . . is the net transport .."

Line 218 Remove "Based on physical principles sea surface slope is proportional to surface geostrophic velocity."

Line 237 Define Tx and Txsw

Line 269 "The coordinates of the along-track altimeter data were obtained from the filtered 12km Jason-2 Aviso satellite product, and not the unfiltered 6 km product which was used for the original ACT proxy [Beal and Elipot, 2016], since the 12 km product matched the ~10 km model resolution more closely." Is this difference significant given that the model is interpolated onto the altimetry ground track?

Figure 2 Caption. Change "Figure 2: HYCOM transport per unit distance proxy (m2 s−1) for Tx (blue) and Txsw (red) transport at 1 km intervals at the first model time step (solid lines, week of 3rd January 1980) and for the mean reference period (dashed lines). The faint grey lines represent the positions of moorings and offshore CPIES pairs." To Figure 2: HYCOM transport per unit distance proxy (m2 s−1) for Tx (blue) and Txsw (red) transport at 1 km intervals at the first model time step (solid lines) and for the ACT reference period (2010-2013, dashed lines). The grey dashed-lines represent the positions of moorings and offshore CPIES pairs."

Line 303-306 remove "Tx and Txsw are simply shown at the first model time step (week of the 3rd of January 1980) in HYCOM and for the mean of the reference period (2010-2013) to show the difference between the net and southwest transport components used to calculate Tbox and Tjet (Figure 2)."

Line 411 Remove "Figure 4 shows the correlation between proxy and model transports for each year."

Line 413 Add "....insignificant minimum correlation of 0.00 (2003) (Figure 4)."

Line 413 Change "... correlation of 0.82 (2014) and an insignificant minimum correlation of 0.00 (2003)." To "... correlation of 0.82 (2014) and a minimum correlation of 0.00 (2003)."

Lines 428-431 Remove. "Figure 5 shows the surface variability by displaying the eddy kinetic energy and the mean surface geostrophic flow as represented by the overlaying

SSH contours over the 3-year reference period, and over the highest (1988) and lowest (1994) correlated years of the box transport proxy."

Any important information in this sentence should be included in the figure caption.

Line 431-432. Add "During the reference period the current appears to be stable with low levels of EKE inshore whereas offshore the flow is more variable with higher levels of EKE (Figure 5)."

Line 445 Remove "Figure 6 shows the mean cross-track velocity profiles during the reference period (2010- 2013), the highest correlated year (1988) and the lowest correlated year (1994) for each mooring and the CPIES-pairs."

Any important information in this sentence should be included in the figure caption.

---

## Editor Comment (EC1) · Hecht (Editor) · 16 Jan 2019

Dear Authors, both reviewers have raised substantial points. If you need more time to respond adequately then please just let us know.

Wishing you the best with the process of revision.

Yours, –Matthew Hecht
* * *

---

## Author Comment (AC2) · 12 Feb 2019

1) The manuscript is very long and would benefit from shortening. There is a lot a duplication between the latter subsections of section 2 and the whole of section 3. Then the authors repeat the results in section 4 again.

Authors- Thank you for this suggestion, the manuscript has been shortened and duplication removed. Changes in manuscript: Sections 1,2,3 & 4

2) The immediate relevance of the study is unclear. The authors present their results very much as a proof-of-concept for the ACT measuring principle, but the moorings

have already been successfully deployed. The motivation therefore feels a bit redundant. Another motivation could be to improve physical understanding of the relationship between SSH and transport, but for that the manuscript is too much focused on the statistics of the relation between the two variables, rather than the hydrodynamics. For example, there are quite a few statements (e.g. line 277 & 279) where a careful analysis of the hydrodynamics would be appropriate

Authors: The goals of the paper were to use the numerical model to test the sensitivity of the transport proxy to i) changes in the vertical structure of the current and how this impacted the linear relationship between SSH slope and transport, and ii) the time period of data needed to build a strong relationship between transport and SSH slope. We appreciate this wasn't clear and have now clarified our goals in the revision. Changes in manuscript: Rephrased this in the Abstract (l 24-26), section 1 (l105-109) as well as in the summary and conclusion section (l430-437).

3) The construction of Tjet and Tbox is quite confusing. For e.g. there is a Tx and a Txsw, even though in both cases they are used for the transport in the southwest (sw) direction. Use better terms for these? Might it help to add the equations how all these transport variables are constructed?

Authors- Txsw is the southwestward component of Tx, we have clarified this in the text. Changes in manuscript: See l 187-191 & l201-210

4) There is no validation of the depth structure of the Agulhas Current in HYCOM. Given that there is quite some mention of the baroclinic nature of the current, this would be good to validate using e.g. the ACT array data themselves.

Authors: Thank you for this suggestion, an important addition to the validation. We have now included a new figure showing the time mean (2010-2013) velocity cross section of the Agulhas Current at the ACT array, for both the ACT in-situ observations and for the HYCOM numerical model. Changes in manuscript: See Fig 2 and l170-176

5) It is a missed opportunity I feel, that the authors have not also investigated the temperature/heat transport. That is something that was hard to do in the ACT array itself, yet is crucial for its climate monitoring ambition. Here, the authors have all the information to calculate the relation between volume and temperature transports

Authors: Unfortunately, this is beyond the scope of the study. The study is focused on investigating the sensitivity of the transport proxy to the underlying assumptions on which it was based. However, it is something we hope to pursue in future.

Other, more minors comments are

- The abstract is fairly technical and detailed, especially in the second half. I am not sure how relevant this is to most readers. For example, how useful is it to mention the terms Tjet and Tbox if they are not explained?

Authors- Noted, the abstract has now been revised. Changes in manuscript: See Abstract

- line 110: add 'time' before 'length scale'? Authors- Noted, however this sentence was removed.

- line 161: It is unclear whether the nesting is one-way or two-way Authors- One way nesting approach and clarified in the text Changes in manuscript: l 130

- Is table 1 really relevant? Most, if not all, of the information is also in the text. And since there is only one model setup, why does it need to be in a table? Authors: We agree and have now removed this table. Changes in manuscript: removed Table 1

- Figure 1: The altimetry line stops just before reaching the shore. Is this an artefact of the plotting, or does this highlight that nearshore altimetry is not used. If the latter, it would be good to mention that Authors- This was the first satellite coordinate point from track 96 (of the TOPEX/Poseidon and Jason satellites) overlapping the starting point of the ACT array

- line 272: I don't understand why the 12km product is used, if the 6km product is more accurate. Why not interpolate the 6km product to the actual mooring locations? Authors- We used the 12km resolution as it more closely matches the 10km resolution of HYCOM. The 6km product also adds more noise/submesoscale processes, which is beyond the resolution of HYCOM to resolve.

- Eq 2: Why not use Tx here, if it is equivalent to Yi?

Authors- Yes, they are equivalent and we have changed Yi to Tx Changes in manuscript: See l268 & Figure 8

- Table 3 would be much more useful if it also listed the observational ACT results?

Authors: We have now included the ACT observational results in Table 2. Changes in manuscript: Table 2

- line 490: Is this increase from 86% to 88% is statistically significant? Authors: Yes, clarified in the manuscript Changes in manuscript: l392

- Table 4: I don't understand why all the r-values are essentially the same. What does this tell us about the system? How to interpret this? And how is the correlation with the observations?

Authors: the performance of the proxy did not necessarily improve by calculating the linear relationship over longer time scales, suggesting that the current dynamics in the model system are very consistent.

Changes in manuscript: See Table 3 & l 499

type-os etc:

- line 62: 'area' instead of 'field'?

- line 120: Zhu et al should be ncitep{}

- line127: 'but may also be'

- line 182: remove 'notably'

- Figure 2: use 'dashed' instead of 'faint'?

- line 641: 'has' instead of 'have'

Authors: Thank you for highlighting these errors, all have now been corrected

Please also note the supplement to this comment:
https://www.ocean-sci-discuss.net/os-2018-117/os-2018-117-AC2-supplement.pdf

[Figure]

**Supplement:**

**Investigating the relationship between volume transport and sea surface height in a numerical ocean model**

Estee Vermeulen [1,2*], Björn Backeberg [2,3,4], Juliet Hermes [1,5], Shane Elipot [6]

[1] Department of Oceanography, University of Cape Town, Rondebosch, South Africa
[2] Nansen-Tutu Centre for Marine Environmental Research, University of Cape Town, South Africa
[3] CSIR, Coastal Systems Research Group, Stellenbosch, South Africa
[4] Nansen Environmental and Remote Sensing Centre, Bergen, Norway
[5] South African Environmental Observation Network, Egagasini Node, Cape Town, South Africa
[6] Rosenstiel School of Marine and Atmospheric Science, University of Miami, 4600 Rickenbacker Causeway, Miami, FL 33149

[*]*Corresponding author address:* Estee Vermeulen, Department of Oceanography, University of Cape Town, Rondebosch, South Africa
Email: esteever01@gmail.com

**Abstract**

The Agulhas Current Time-series mooring array (ACT) measured transport of the Agulhas Current at 34°S for a period of 3 years. Using along-track satellite altimetry data directly above the array, a proxy of Agulhas Current transport was developed based on the relationship between cross-current sea surface height (SSH) gradients and the measured transports. In this study, the robustness of the proxy is tested within a numerical modelling framework, using a 34-year long regional-hindcast simulation from the Hybrid Coordinate Ocean Model (HYCOM). The model specifically tested the sensitivity of the transport proxy to (1) changes in the vertical structure of the current and to (2) different sampling periods used to calculate the proxy. Two reference proxies were created using HYCOM data from 2010-2013, by extracting model data at the mooring positions and along the satellite altimeter track for; the box (net) transport and the jet (southwestward) transport. Sensitivity tests were performed where the proxy was recalculated from HYCOM for (1) a period where the modelled vertical stratification was different compared to the reference proxy, and (2) different lengths of time periods: 1, 3, 6, 12, 18 and 34 years. Compared to the simulated (native) transports, it was found that the HYCOM proxy was more capable of estimating the box transport of the Agulhas Current compared to the jet transport. This was because the model is unable to resolve the dynamics associated with meander events, for which the jet algorithm was developed. The HYCOM configuration in this study contained exaggerated levels of offshore variability in the form of frequently-impinging baroclinic anticyclonic eddies. These eddies consequently broke down the linear relationship between SSH slope and vertically-integrated transport. Lastly, results showed that calculating the proxy over shorter or longer time periods in the model did not significantly impact the skill of the Agulhas transport proxy, suggesting that 3-years was a sufficiently long time-period for the observation based transport proxy. These results were consistent to a previous study that was used to design the ACT mooring array and therefore supports research methods needed to develop future monitoring programs of the Agulhas Current System.

[revised manuscript text omitted]

---

## Author Comment (AC3) · 12 Feb 2019

The manuscript would benefit with a reorder and editing. For example, in the Summary and Conclusion section the authors state (line 534-537) "The HYCOM model provided the means to investigate the validity of the assumptions used to create the proxies, such as the constant relationship between SSH slope and transport per unit distance at each mooring location and the temporal scale of observations needed to build a strong linear relationship between transport and SSH slope." They then follow with a limited discussion explaining some reasons why the proxy does not capture the model transport, referring to figures to justify this reasoning – this is not a summary

or conclusion. It is suggested that much of the information (lines 534-628) should be incorporated into the relevant parts of Section 3. A reordering of Lines 629-696 would form what may be considered a "Summary and Conclusion" sections.

Authors- Thank you for these suggested changes which were in line with the other reviewer. We have now significantly condensed and clarified the text and flow of the paper.

Section 2.1 should only provide details of the model used in this study. The reader is not interested in the details of the larger regional model that provided the boundary conditions of the higher resolution (1/10o) nested model.

Authors: Noted and changed Changes in manuscript: see section 2.1

The presentation of section 2 was convoluted and thus difficult for the reader to easily understand the approach taken. It is suggested that the authors revise this section to more clearly and concisely explaining the methods and assumptions.

Authors: Noted and revised Changes in manuscript: Section 2

Lines 275-290 "The length scales of the slopes ranged from 24 km at mooring A to 12 km at mooring G and 48 km for the offshore CPIES-pairs, indicating an increase in the spatial scale of offshore flow, possibly due to increased offshore variability. Results from the in situ proxy experiment by Beal and Elipot [2016] also showed an increasing length scale with increasing distance offshore, however the results varied considerably in magnitude: 27 km at mooring B to 102 km at mooring G." Can you explain the reason for the difference in length scales between the model and observations (in situ and satellite)? Does this indicate the model doesn't capture the observed variability? What implications does this have for this study?

Authors: The reason why the length scales differ between the model and the observations is because the model does not capture completely and accurately the observed variability. This limitation and its implication is now discussed in this study and clarified

in the text. Changes in manuscript: See l 440

It is suggested that section 2.4.1 be revised to remove any unnecessary information concerning the larger regional model.

Authors: Noted and changed Changes in manuscript: l193

Line 407-408. "The proxies only capture a portion of the transport estimate from the HYCOM model, suggesting it also only captures a portion of the model variability." Is this the only problem with the proxy estimate? A more detailed analysis is really required to understand the impact of the assumptions used in developing the proxy.

Authors: The frequently impinging eddies make it difficult for the proxy to accurately estimate the transport of both Tbox and Tjet because the eddies resulted in the correlation of the regression models decreasing offshore. Therefore, the proxy transport estimates (for both Tbox and Tjet) inshore were more accurate than the ones offshore. We have clarified this in the text Changes in manuscript: l 439-470.

Line 418-420 "In summary, the results indicate that the proxy is generally better suited in HYCOM to estimate the box transport rather than the jet transport. Further analysis in this study therefore only focuses on the box transport." It is not appropriate to simply ignore results that don't agree. You need to fully explore the reasons why the different proxies fail.

Authors: The difference in the performance of the jet transport algorithm in the models and in the observations suggests that the models are unable to resolve all the dynamics associated with meander events, for which the jet algorithm was specifically developed. The jet transport proxy by Beal and Elipot [2016] was developed to estimate the transport of the Agulhas Current during mesoscale meander events, which generally causes the current to manifest as a full-depth, surface intensified, cyclonic circulation out to 150 km from the coast with anticyclonic circulation farther offshore. The Agulhas meanders in the HYCOM simulation occur in association with large anticyclonic eddies

predominantly located at the offshore edge of the current, with a narrow, southwest stream close to the coast. In some instances anticyclonic eddies span the length of the entire array. Therefore, considering that the model is unable to resolve the dynamics associated with meander events, for which the jet transport algorithm was specifically developed, further analysis only focuses on the box transport proxy In addition, the poorer performance of the Tjet proxy in HYCOM and possibly in the in situ study, may also be because it only represents the southwestward component of the flow, whereas the input sea surface slope reflects the net flow along the array. Therefore, based on these findings further analysis focussed on the Tbox proxy. We have explored this in the text. Changes in manuscript: see l33, l335-344 & l438-461

Lines 485-499 Removing outlier to increase the performance of the proxy is not appropriate. The authors should clearly identify the dynamical reasons for the reduced skill of the proxy. It is only through this in-depth analysis that advantages and disadvantages of the proxy can be fully explored.

Authors- The reason we decided to remove the outliers was because in the case of the offshore linear models, the outliers were often the transport events that violated the linear relationship between SSH slope and transport. Investigation into the current structure of the outlying transport events further showed the baroclinic nature of the eddies that broke down the linear relationship between SSH slope and transport, specifically for the offshore regression models. Thus, removing the transport events that violated the relationship proved to increase the performance of the proxy. Motivating that the offshore variability resulted in the poorer performance of the models offshore. Changes in manuscript: see l390-l395 & l471-474 & Figure 4

The manuscript is lengthy and the prose overly convoluted and repetitive, when reviewing the manuscript the authors should, where possible, simplify the writing and remove repetition. Below are a few examples:

Authors: Thank you for highlighting this, we have thoroughly revised the manuscript to

improve readability.

Line 85-89 "The Agulhas transport proxy of Beal and Elipot [2016] was derived from the physical principle of geostrophy, where along-track sea surface height slope measured by satellite altimeters can ultimately be related to a measure of volume transport across a portion of the current, provided that the surface current represents the flow at depth [Beal and Elipot, 2016]. " can be deleted as lines 89-93 fully explain the major findings of the Beal and Elipot, 2016 study.

Authors: Noted and corrected

Line 151 change ": : : in doing so: : :" to ".. thus : : :"

Authors: Corrected

Line159-161 remove " The horizontal resolution of the parent model ranged from 14 km in the northern Indian Ocean to 45 km in the Southern Ocean, with a resolution ranging from 30 to 40 km in the region of the Agulhas Current." This information is not needed; the reader can refer to George et al., 2010 if they require more information on the model from which the boundary conditions were taken.

Authors: Noted and corrected

Line 154-155 Change "The HYCOM output in this study was made available from a nested 1/10_ model of the greater Agulhas Current System (AGULHAS) [Backeberg et al., 2008; 2009; 2014]." To "This study used output from a nested 1/10_ model of the greater Agulhas Current System (AGULHAS) [Backeberg et al., 2008; 2009; 2014]."

Authors: Noted and corrected

These are a few examples; there are many more instances of repetition and where more concise writing would improve the text. Minor comments:

Line 45 change "As the current continues southwestward the current becomes.." to "As the current continues southwestward it becomes.. "

Authors: Noted and corrected

Line 60-62 poorly constructed sentence "The unique circulation of the Agulhas Current System, in the context of regional and global climates, makes it an important field of research."

Authors: The unique circulation of the ACS, in the context of regional and global climate variability, makes it an important field of research Changes to manuscript- l63

Line 67: "However, the close proximity of the current to the coast makes it difficult to monitor using satellite altimetry [Rouault et al., 2010]." Is this statement still true given the development of the AVISO X-track product (https://www.aviso.altimetry.fr/en/data/products/sea-surface-heightproducts/ regional/x-track-sla/coastal-along-track-sea-level-anomalies.html)?

Authors: Noted and addressed Changes in manuscript: in l70-73. The close proximity of the Agulhas Current to the coast has made it difficult to monitor using satellite altimetry, however, newer altimetry products dedicated to coastal areas are promising but are yet to be validated within the Agulhas Current region (Birol et al., 2017).

Line 74-84. It can be shown that the total cost of in situ observing, satellite observations and models are all on similar cost. Singling out in situ observations as the only costly tool is not appropriate or accurate.

Authors: Noted, cost aspect removed. Changes in manuscript: l80

Change " In situ observations may accurately measure the dynamics of the Agulhas Current throughout the water column but are expensive and spatially coarse." To "In situ mooring observations provide high temporal observations of the Agulhas Current throughout the water column but spatially coarse."

Authors: Noted and corrected

Line 106 Change [Beal and Elipot, 2016] to Beal and Elipot [2016] Authors: Noted and

corrected

Line 120 Change Zhu et al. [2004] to [Zhu et al., 2004] Authors: Noted and corrected

Line 158-159 Change ".. buffer zone." To ".. sponge layer." Authors: Noted and corrected

Line 166-167 Change "Both models have 30 hybrid layers and targeted densities ranging from 23.6 to 27.6 kg/m3. To "AGULHAS has 30 hybrid layers and targeted densities ranging from 23.6 to 27.6 kg/m3." Authors: Noted and corrected

Line 185 Add " : : : 2010-2013 (Figure 1, Beal et al., 2015) Authors: Noted and corrected

Line 193-195 Change "During the first phase of the ACT experiment, the mooring array was maintained in the Agulhas Current for a period of 34 months, perpendicular to the continental slope at 34_S, south of East London, South Africa (Figure 1)." To " The ACT mooring array was located perpendicular to the continental slope at 34_S, south of East London, South Africa (Figure 1)." Authors: Noted and corrected

Line 200 Change " From the data collected in Beal et al. [2015], two volume transports were estimated:: : : " to "From the data collected, Beal et al. [2015], provided two volume transports estimates: .." Authors: Noted and corrected

Line 202 Change ": : : is a net transport" to ": : : is the net transport .." Authors: noted and corrected

Line 218 Remove "Based on physical principles sea surface slope is proportional to surface geostrophic velocity." Authors: Removed

Line 237 Define Tx and Txsw

Authors: The transport variable in the regression models was defined as transport per unit distance, i.e. the vertically integrated velocity with units in ms2.s-1 where Tx represents the net component of the current flow and Txsw the southwestward

component of the flow. Changes in manuscript: See l187-191

Line 269 "The coordinates of the along-track altimeter data were obtained from the filtered 12km Jason-2 Aviso satellite product, and not the unfiltered 6 km product which was used for the original ACT proxy [Beal and Elipot, 2016], since the 12 km product matched the _10 km model resolution more closely." Is this difference significant given that the model is interpolated onto the altimetry ground track?

Authors: No, the difference is not significant. However, at the time, we decided to use the 12km resolution as it more closely matches the 10km resolution of HYCOM.

Figure 2 Caption. Change "Figure 2: HYCOM transport per unit distance proxy (m2 sôĂĂĂ1) for Tx (blue) and Txsw (red) transport at 1 km intervals at the first model time step (solid lines, week of 3rd January 1980) and for the mean reference period (dashed lines). The faint grey lines represent the positions of moorings and offshore CPIES pairs." To Figure 2: HYCOM transport per unit distance proxy (m2 sôĂĂĂ1) for Tx (blue) and Txsw (red) transport at 1 km intervals at the first model time step (solid lines) and for the ACT reference period (2010-2013, dashed lines). The grey dashed-lines represent the positions of moorings and offshore CPIES pairs." Authors: Noted and corrected

Line 303-306 remove "Tx and Txsw are simply shown at the first model time step (week of the 3rd of January 1980) in HYCOM and for the mean of the reference period (2010-2013) to show the difference between the net and southwest transport components used to calculate Tbox and Tjet (Figure 2)." Authors: Removed

Line 411 Remove "Figure 4 shows the correlation between proxy and model transports for each year." Authors: Removed

Line 413 Add ": : :.insignificant minimum correlation of 0.00 (2003) (Figure 4)." Authors: Noted

Line 413 Change ": : : correlation of 0.82 (2014) and an insignificant minimum correlation of 0.00 (2003)." To ": : : correlation of 0.82 (2014) and a minimum correlation of 0.00 (2003)." Authors: Noted

Lines 428-431 Remove. "Figure 5 shows the surface variability by displaying the eddy kinetic energy and the mean surface geostrophic flow as represented by the overlaying SSH contours over the 3-year reference period, and over the highest (1988) and lowest (1994) correlated years of the box transport proxy."

Any important information in this sentence should be included in the figure caption. Authors: Noted and Removed

Line 431-432. Add "During the reference period the current appears to be stable with low levels of EKE inshore whereas offshore the flow is more variable with higher levels of EKE (Figure 5)." Authors: Noted

Line 445 Remove "Figure 6 shows the mean cross-track velocity profiles during the reference period (2010- 2013), the highest correlated year (1988) and the lowest correlated year (1994) for each mooring and the CPIES-pairs." Any important information in this sentence should be included in the figure caption Authors: Noted and Removed

Please also note the supplement to this comment:
https://www.ocean-sci-discuss.net/os-2018-117/os-2018-117-AC3-supplement.pdf